# Ozone and haze pollution weakens net primary productivity in China

Xu Yue[1], Nadine Unger[2], Kandice Harper[3], Xiangao Xia[4], Hong Liao[5], Tong Zhu[6], Jingfeng Xiao[7], Zhaozhong Feng[8], and Jing Li[9]

[1] Climate Change Research Center, Institute of Atmospheric Physics, Chinese Academy of Sciences, Beijing 100029, China
[2] College of Engineering, Mathematics and Physical Sciences, University of Exeter, Exeter, EX4 4QE, UK
[3] School of Forestry and Environmental Studies, Yale University, 195 Prospect Street, New Haven, Connecticut 06511, USA
[4] Laboratory for Middle Atmosphere and Global Environment Observation, Institute of Atmospheric Physics, Chinese Academy of Sciences, Beijing 100029, China
[5] School of Environmental Science and Engineering, Nanjing University of Information Science & Technology, Nanjing 210044, China
[6] State Key Laboratory for Environmental Simulation and Pollution Control, College of Environmental Sciences and Engineering, Peking University, Beijing 100871, China.
[7] Earth Systems Research Center, Institute for the Study of Earth, Oceans, and Space, University of New Hampshire, Durham, NH 03824, USA
[8] Research Center for Eco-Environmental Sciences, Chinese Academy of Sciences, Beijing 100085, China
[9] Laboratory for Climate and Ocean-Atmosphere Studies, Department of Atmospheric and Oceanic Sciences, School of Physics, Peking University, Beijing 100871, China

*Corresponding author:*
Xu Yue
Telephone: 86-10-82995369
Email: xuyueseas@gmail.com

*Keywords:* Haze pollution, climate projection, pollution mitigation, ozone damage, diffuse radiative fertilization, aerosol radiative effects, aerosol indirect effects, photosynthesis, net primary productivity

**Abstract**

Atmospheric pollutants have both beneficial and detrimental effects on carbon uptake by land ecosystems. Surface ozone ($O_3$) damages leaf photosynthesis by oxidizing plant cells, while aerosols promote carbon uptake by increasing diffuse radiation and exert additional influences through concomitant perturbations to meteorology and hydrology. China is currently the world's largest emitter of both carbon dioxide and short-lived air pollutants. The land ecosystems of China are estimated to provide a carbon sink, but it remains unclear whether air pollution acts to inhibit or promote carbon uptake. Here, we employ Earth system modeling and multiple measurement datasets to assess the separate and combined effects of anthropogenic $O_3$ and aerosol pollution on net primary productivity (NPP) in China. In the present day, $O_3$ reduces annual NPP by 0.6 Pg C (14%) with a range from 0.4 Pg C (low $O_3$ sensitivity) to 0.8 Pg C (high $O_3$ sensitivity). In contrast, aerosol direct effects increase NPP by 0.2 Pg C (5%) through the combination of diffuse radiation fertilization, reduced canopy temperatures, and reduced evaporation leading to higher soil moisture. Consequently, the net effects of $O_3$ and aerosols decrease NPP by 0.4 Pg C (9%) with a range from 0.2 Pg C (low $O_3$ sensitivity) to 0.6 Pg C (high $O_3$ sensitivity). However, precipitation inhibition from combined aerosol direct and indirect effects reduces annual NPP by 0.2 Pg C (4%), leading to a net air pollution suppression of 0.8 Pg C (16%) with a range from 0.6 Pg C (low $O_3$ sensitivity) to 1.0 Pg C (high $O_3$ sensitivity). Our results reveal strong dampening effects of air pollution on the land carbon uptake in China today. Following the current legislation emission scenario, this suppression will be further increased by the year 2030, mainly due to a continuing increase in surface $O_3$. However, the maximum technically feasible reduction scenario could drastically relieve the current level of NPP damage by 70% in 2030, offering protection of this critical ecosystem service and the mitigation of long-term global warming.

# 1 Introduction

Surface ozone ($O_3$) and atmospheric aerosols influence land ecosystem carbon uptake both directly and indirectly through Earth system interactions. $O_3$ reduces plant photosynthesis directly through stomatal uptake. The level of damage is dependent on both surface ozone concentrations ($[O_3]$) and the stomatal conductance ($g_s$), the latter of which is closely related to the photosynthetic rate (Reich and Amundson, 1985; Sitch et al., 2007; Ainsworth et al., 2012). The impact of aerosol pollution on vegetation is less clear. Atmospheric aerosols influence plant photosynthesis through perturbations to radiation, meteorology, and clouds. Observations (Cirino et al., 2014; Strada et al., 2015) suggest that an increase in diffuse light partitioning in response to a moderate aerosol loading can improve canopy light use efficiency (LUE) and promote photosynthesis, known as diffuse radiation fertilization (DRF), as long as the total light availability is not compromised (Kanniah et al., 2012). Atmospheric aerosols also reduce leaf temperature (Steiner and Chameides, 2005; Cirino et al., 2014), but the consequence for photosynthesis depends on the relationship between the local environmental temperature and the photosynthetic optimum temperature of approximately 25°C. Aerosol-induced changes in evaporation and precipitation are interconnected but impose opposite effects on photosynthesis; less evaporation preserves soil moisture in the short term but may decrease local rainfall (Spracklen et al., 2012) and lead to drought conditions in the long term. Furthermore, aerosol indirect effects (AIE) on cloud properties can either exacerbate or alleviate the above feedbacks.

China is currently the world's largest emitter of both carbon dioxide and short-lived air pollutants (http://gains.iiasa.ac.at/models/). The land ecosystems of China are estimated to provide a carbon sink (Piao et al., 2009), but it remains unclear how air pollution may affect this sink through the atmospheric influences on regional carbon uptake. $O_3$ damages to photosynthesis, including those in China, have been quantified in hundreds of measurements (Table S1), but are limited to individual plant species and specific $O_3$ concentrations ($[O_3]$). Previous regional modeling of $O_3$ vegetation damage (e.g., Ren et al., 2011; Tian et al., 2011) does not always take advantage of valuable observations to

calibrate GPP-$O_3$ sensitivity coefficients for China domain and typically the derived
results have not been properly validated. The aerosol effects on photosynthesis are less
well understood. Most of the limited observation-based studies (Rocha et al., 2004;
Cirino et al., 2014; Strada et al., 2015) rely on long-term flux measurements or satellite
retrievals, which are unable to unravel impacts of changes in the associated
meteorological and hydrological forcings. Modeling studies focus mainly on the aerosol-
induced enhancement in diffuse radiation (e.g., Cohan et al., 2002; Gu et al., 2003;
Mercado et al., 2009), but ignore other direct and indirect feedbacks such as changes in
temperature and precipitation. Finally, no studies have investigated the combined effects
of $O_3$ and aerosols or how the air pollution influences may vary in response to future
emission regulations and climate change.

In this study, we assess the impacts of $O_3$ and aerosols on land carbon uptake in China
using the global Earth system model NASA GISS ModelE2 that embeds the Yale
Interactive Terrestrial Biosphere model (YIBs). This framework is known as NASA
ModelE2-YIBs and fully couples the land carbon-oxidant-aerosol-climate system
(Schmidt et al., 2014; Yue and Unger, 2015). The global-scale model accounts for long-
range transport of pollution and large-scale feedbacks in physical climate change. The
coupled Earth system simulations apply present-day and future pollution emission
inventories from the Greenhouse Gas and Air Pollution Interactions and Synergies
(GAINS) integrated assessment model (http://gains.iiasa.ac.at/models/). The simulations
include process-based, mechanistic photosynthetic responses to physical climate change,
$O_3$ stomatal uptake, carbon dioxide ($CO_2$) fertilization, and aerosol radiative
perturbations, but not aerosol and acid deposition (Table 1). The $O_3$ and aerosol haze
effects on the land carbon cycle fluxes occur predominantly through changes to gross
primary productivity (GPP) and net primary productivity (NPP). Therefore, the current
study focuses on GPP and NPP impacts and does not address changes in net ecosystem
exchange (NEE).

**2 Methods**

## 2.1 YIBs vegetation model

**2.1 YIBs vegetation model**

The YIBs model applies the well-established Farquhar and Ball-Berry models (Farquhar
et al., 1980; Ball et al., 1987) to calculate leaf photosynthesis and stomatal conductance,
and adopts a canopy radiation scheme (Spitters, 1986) to separate diffuse and direct light
for sunlit and shaded leaves. The assimilated carbon is dynamically allocated and stored
to support leaf development (changes in leaf area index, LAI) and tree growth (changes
in height). A process-based soil respiration scheme that considers carbon flows among 12
biogeochemical pools is included to simulate carbon exchange for the whole ecosystem
(Yue and Unger, 2015). Similar to many terrestrial models (Schaefer et al., 2012), the
current version of YIBs does not include a dynamic N cycle. Except for this deficit, the
vegetation model can reasonably simulate ecosystem responses to changes in $[CO_2]$,
meteorology, phenology, and land cover (Yue et al., 2015). A semi-mechanistic $O_3$
vegetation damage scheme (Sitch et al., 2007) is implemented to quantify responses of
photosynthesis and stomatal conductance to $O_3$ (Yue and Unger, 2014).

The YIBs model can be used in three different configurations: site-level, global/regional
offline, and online within ModelE2-YIBs (Yue and Unger, 2015). The offline version is
driven with hourly 1°×1° meteorological forcings from either the NASA Modern Era
Retrospective-analysis for Research and Applications (MERRA) (Rienecker et al., 2011)
or the interpolated output from ModelE2-YIBs. The online YIBs model is coupled with
the climate model NASA ModelE2 (Schmidt et al., 2014), which considers the interplay
among meteorology, radiation, atmospheric chemistry, and plant photosynthesis at each
time step. For both global and regional simulations, 8 plant functional types (PFTs) are
considered (Fig. S1). This land cover is aggregated from a dataset with 16 PFTs, which
are derived using retrievals from both the Moderate Resolution Imaging
Spectroradiometer (MODIS) (Hansen et al., 2003) and the Advanced Very High
Resolution Radiometer (AVHRR) (Defries et al., 2000). The same vegetation cover with
16 PFTs is used by the Community Land Model (CLM) (Oleson et al., 2010).

Both the online and offline YIBs models have been extensively evaluated with site-level
measurements from 145 globally-dispersed flux tower sites, long-term gridded
benchmark products, and multiple satellite retrievals of LAI, tree height, phenology, and
carbon fluxes (Yue and Unger, 2015; Yue et al., 2015). Driven with meteorological
reanalyses, the offline YIBs vegetation model estimates a global GPP of $122.3 \pm 3.1$ Pg C
$yr^{-1}$, NPP of $63.6 \pm 1.9$ Pg C $yr^{-1}$, and NEE of -2.4 $\pm$ 0.7 Pg C $yr^{-1}$ for 1980-2011,
consistent with an ensemble of land models (Yue and Unger, 2015). The online
simulations with ModelE2-YIBs, including both aerosol effects and $O_3$ damage, yield a
global GPP of $125.8 \pm 3.1$ Pg C $yr^{-1}$, NPP of $63.2 \pm 0.4$ Pg C $yr^{-1}$, and NEE of -3.0 $\pm$ 0.4
Pg C $yr^{-1}$ under present day conditions.

**2.2 NASA ModelE2-YIBs model**

The NASA ModelE2-YIBs is a fully coupled chemistry-carbon-climate model with
horizontal resolution of $2° \times 2.5°$ latitude by longitude and 40 vertical levels extending to
0.1 hPa. The model simulates gas-phase chemistry ($NO_x$, $HO_x$, $O_x$, CO, $CH_4$, NMVOCs),
aerosols (sulfate, nitrate, elemental and organic carbon, dust, sea salt), and their
interactions (Schmidt et al., 2014). Modeled oxidants influence the photochemical
formation of secondary aerosol species (sulfate, nitrate, secondary organic aerosol). In
turn, modeled aerosols affect photolysis rates in the online gas-phase chemistry (Schmidt
et al., 2014). Heterogeneous chemistry on dust surfaces is represented (Bauer et al.,
2007). The embedded radiation package includes both direct and indirect (Menon and
Rotstayn, 2006) radiative effects of aerosols and considers absorption by multiple GHGs.
Size-dependent optical parameters of clouds and aerosols are computed from Mie
scattering, ray tracing, and T-matrix theory, and include the effects of non-spherical
particles for cirrus and dust (Schmidt et al., 2006). Simulated surface solar radiation
exhibits the lowest model-to-observation biases compared with the other 20 IPCC-class
climate models (Wild et al., 2013). Simulated meteorological and hydrological variables
have been full validated against observations and reanalysis products (Schmidt et al.,

186 2014).


**2.3 Emissions**

We use global annual anthropogenic pollution inventories from the GAINS integrated
assessment model (Amann et al., 2011), which compiles historic emissions of air
pollutants for each country based on available international emission inventories and
national information from individual countries. Inter-comparison of present-day (the year
2010) emissions (Fig. S2) shows that the GAINS V4a inventory has similar emission
intensity (within ±10%) in China as IPCC RCP8.5 scenario (van Vuuren et al., 2011) for
most species, except for ammonia, which is higher by 80% in GAINS. The discrepancies
among different inventories emerge from varied assumptions on the stringency and
effectiveness of emission control measures. While the GAINS 2010 ammonia emissions
from China are larger than the RCP8.5 and HTAP emissions as shown in Fig. S2, they
are close in magnitude to the year 2010 emissions of 13.84 Tg yr$^{-1}$ estimated by the
Regional Emission inventory in ASia (REAS, http://www.nies.go.jp/REAS/).

The GAINS inventory also projects medium-term variations of future emissions at five-
year intervals to the year 2030. The current legislation emissions (CLE) scenario applies
full implementation of national legislation affecting air pollution emissions; for China,
this represents the 11$^{th}$ five-year plan, including known failures. By 2030, in the CLE
inventory, CO decreases by 18%, $SO_2$ by 21%, black carbon (BC) by 28%, and organic
carbon (OC) by 41%, but $NO_x$ increases by 20%, ammonia by 22%, and non-methane
volatile organic compounds (NMVOC) by 6%, relative to the 2010 emission magnitude
in China. To account for potential rapid changes in policy and legislation, we apply the
maximum technically feasible reduction (MTFR) emission scenario as the lower limit of
future air pollution. The MTFR scenario implements all currently available control
technologies, disregarding implementation barriers and costs. With this scenario, the
2030 emissions of $NO_x$ decrease by 76%, CO by 79%, $SO_2$ by 67%, BC by 81%, OC by
89%, ammonia by 65%, and NMVOC by 62% in China, indicating large improvement of
air quality. Biomass burning emissions, adopted from the IPCC RCP8.5 scenario (van
Vuuren et al., 2011), are considered as anthropogenic sources because most fire activities
in China are due to human-managed prescribed burning (Zhou et al., 2017). Compared
with the GAINs inventory, present-day biomass burning is equivalent to <1% of the
emissions for $NO_x$, $SO_2$, and $NH_3$, 1.6% for BC, 3.0% for CO, and 9.6% for OC. By the
year 2030, biomass burning emissions decrease by 1-2% for all pollution species
compared with 2010.

The model represents climate-sensitive natural precursor emissions of lightning $NO_x$, soil
$NO_x$ and biogenic volatile organic compounds (BVOCs) (Unger and Yue, 2014). Future
2030 changes in these natural emissions are small compared to the anthropogenic
emission changes. Interactive lightning $NO_x$ emissions are calculated based on the
climate model's moist convection scheme that is used to derive the total lightning and the
cloud-to-ground lightning frequencies (Price et al., 1997; Pickering et al., 1998; Shindell
et al., 2013). Annual average lightning $NO_x$ emissions over China increase by 4%
between 2010 and 2030. Interactive biogenic soil $NO_x$ emission is parameterized as a
function of PFT-type, soil temperature, precipitation (including pulsing events), fertilizer
loss, LAI, $NO_x$ dry deposition rate, and canopy wind speed (Yienger and Levy, 1995).
Annual average biogenic soil $NO_x$ emissions increase by only 1% over China between
2010 and 2030. Leaf isoprene emissions are simulated using a biochemical model that
depends on the electron transport-limited photosynthetic rate, intercellular $CO_2$, canopy
temperature, and atmospheric $CO_2$ (Unger et al., 2013). Leaf monoterpene emissions
depend on canopy temperature and atmospheric $CO_2$ (Unger and Yue, 2014). Annual
average isoprene emission in China increases by 5% (0.39 Tg C $yr^{-1}$) between 2010 and
2030 in response to enhanced GPP and temperature that offset the effects of $CO_2$-
inhibition. Monoterpene emissions decrease by 5% (-0.25 Tg C) between 2010 and 2030
because $CO_2$-inhibition outweighs the effects of increased warming.

**2.4 Simulations**

**2.4.1 NASA ModelE2-YIBs online**
We perform 24 time-slice simulations to explore the interactive impacts of $O_3$ and
aerosols on land carbon uptake (Table 2). All simulations are performed in atmosphere-
only configuration. In these experiments, [$O_3$] and aerosol loading are dynamically
predicted, and atmospheric chemistry processes are fully two-way coupled to the
meteorology and the land biosphere. Simulations can be divided into two groups,
depending on whether AIE are included. In each group, three subgroups are defined with
the emission inventories of GAINS 2010, CLE 2030, and MTFR 2030 scenarios. In each
subgroup, one baseline experiment is set up with only natural emissions (denoted with
NAT). The other three implement all natural and anthropogenic sources of emissions
(denoted with ALL), but apply different levels of $O_3$ damage including none (denoted
with NO3), low sensitivity (LO3), and high sensitivity (HO3). To compare the
differences between online and offline $O_3$ damage, we perform four additional
simulations which do not account for the feedbacks of $O_3$-induced changes in
biometeorology, plant growth, and ecosystem physiology. Two simulations,
G10ALLHO3_OFF and G10ALLLO3_OFF, include both natural and anthropogenic
emissions. The other two, G10NATHO3_OFF and G10NATLO3_OFF, include natural
emissions alone.

We use prescribed sea surface temperature (SST) and sea ice distributions simulated by
ModelE2 under the IPCC RCP8.5 scenario (van Vuuren et al., 2011). For these boundary
conditions, we apply the monthly-varying decadal average of 2006-2015 for 2010
simulations and that of 2026-2035 for 2030 simulations. Well-mixed GHG
concentrations are also adopted from the RCP8.5 scenario, with $CO_2$ changes from 390
ppm in 2010 to 449 ppm in 2030, and $CH_4$ changes from 1.779 ppm to 2.132 ppm. Land
cover change projections for this scenario suggest only minor changes between the years
2010 and 2030; for example, the expansion of 3% for grassland is offset by the losses of
1% for cropland and 4% for tropical rainforest. As a result, we elect to apply the same
land cover, which is derived from satellite retrievals, for both present-day and future
simulations (Fig. S1). We use present-day equilibrium tree height derived from a 30-year
spinup procedure (Yue and Unger, 2015) as the initial condition. All simulations are
performed for 20 years, and the last 15 years are used for analyses. For simulations
including effects of $CO_2$ fertilization, climate change, and $O_3$ damages, GPP and NPP
reach new equilibrium within 5 years while those for NEE may require several decades
due to the slow responses of the soil carbon pools (Fig. S3). The full list of simulations in
Table 2 offers assessment of uncertainties due to interannual climate variability, emission
inventories (CLE or MTFR), $O_3$ damage sensitivities (low to high), and aerosol climatic
effects (direct and indirect). Uncertainties calculated based on the interannual climate
variability in the model are indicated using the format 'mean ± one standard deviation'.
Other sources of uncertainty are explicitly stated.

**2.4.2 YIBs offline with MERRA meteorology**
We perform 15 simulations to evaluate the skill of the $O_3$ damage scheme for vegetation
in China (Table S2). Each run is driven with hourly meteorological forcings from NASA
GMAO MERRA (Rienecker et al., 2011). One baseline simulation is performed without
inclusion of any $O_3$ damage. The others, seven runs in each of two groups, are driven
with fixed [$O_3$] at 20, 40, 60, 80, 100, 120, and 140 ppbv, respectively, using either low
or high $O_3$ sensitivities defined by (Sitch et al., 2007). Thus, [$O_3$] in these offline runs is
fixed without seasonal and diurnal variations to mimic field experiments that usually
apply a constant level of [$O_3$] during the test period. We compare the $O_3$-affected GPP
with the $O_3$-free GPP from the baseline simulation to derive the damaging percentages to
GPP, which are compared with values for different PFTs from an ensemble of published
literature results (Table S1). All simulations are performed for 1995-2011, and the last 10
years are used for analyses.

**2.4.3 YIBs offline with ModelE2-YIBs meteorology**
Using the offline YIBs vegetation model driven with ModelE2-YIBs meteorology, we
perform 30 simulations to isolate the impacts of aerosol-induced changes in the
individual meteorological drivers on carbon uptake (Table S3). Experiments are
categorized into two groups, depending on whether the GCM forcings include AIE or
not. In each group, three subgroups of simulations are set up with different meteorology
for GAINS 2010, CLE 2030, and MTFR 2030 scenarios. Within each subgroup, five runs
are designed with the different combinations of GCM forcings. One baseline run is forced
with meteorology simulated without anthropogenic aerosols. The other four are
additionally driven with aerosol-induced perturbations in temperature, PAR, soil
moisture, or the combination of the above three variables. For these simulations, the
month-to-month meteorological perturbations caused by aerosols are applied as scaling
factors on the baseline forcing for each month at each grid square. The differences of
NPP between sensitivity and baseline runs represent contributions of individual or total
aerosol effects. Each simulation is performed for 15 years, with the last 10 years used for
analyses. Uncertainties due to interannual climate variability in the model are calculated
using different time periods for the online (15 years, Table 2) and offline (10 years, Table
S3) runs.

**3  Results**

**3.1 Evaluation of ModelE2-YIBs over China**

**3.1.1 Land carbon fluxes: GPP and NPP**
To validate simulated GPP, we use a gridded benchmark product for 2009-2011 upscaled
from *in situ* FLUXNET measurements (Jung et al., 2009). For NPP, we use a MODIS
satellite-derived product for 2009-2011 (Zhao et al., 2005). Both datasets are interpolated
to the same resolution of $2°×2.5°$ as ModelE2-YIBs. Simulated GPP and NPP reproduce
the observed spatial patterns with high correlation coefficients (R=0.46-0.86, $p < 0.001$)
and relatively low model-to-observation biases (< 21% on national scale) (Fig. 1 and Fig.
S4). High values of the land carbon fluxes are predicted in the East and the Northeast,
where forests and croplands dominate (Fig. S1). For GPP, prediction in the summer
overestimates by 6.2% over the southern coast (< 28°N), but underestimates by 23.7%
over the North China Plain (NCP, [32-40°N, 110-120°E]). Compared with the MODIS
data product, predicted summer NPP is overall overestimated by 20.6% in China (Fig.
1f), with regional biases of 40.0% in the southern coast, 51.2% in the NCP, and 38.7% in
the Northeast (> 124°E).

**3.1.2 Surface air pollution and AOD**
For surface concentrations of $PM_{2.5}$ and $O_3$, we use ground measurements available for
2014 from 188 sites operated by the Ministry of Environmental Protection of China
(http://www.aqicn.org/). In addition, we use rural [$O_3$] from published literature (Table
S4) to evaluate the model. For AOD, we use gridded observations of 2008-2012 from
MODIS retrievals. The model simulates reasonable magnitude and spatial distribution of
surface $PM_{2.5}$ concentrations (Fig. 2 and Fig. S5). Predicted AOD also reproduces the
observed spatial pattern, but underestimates the high center in NCP by 24.6% in summer.
Long-term measurements of $[O_3]$ are very limited in China. Comparisons with the 2014
one-year data from 188 urban sites show that simulated $[O_3]$ reproduces reasonable
spatial distribution but overestimates the average concentration by >40% (Fig. 2f and Fig.
S5f). Such discrepancy is in part attributed to the sampling biases, because urban $[O_3]$ is
likely lower than rural $[O_3]$ due to high $NO_x$ emissions ($NO_x$ titration) and aerosol loading
(light extinction) in cities. Based on 'China Statistical Yearbook for 2015'
(http://www.stats.gov.cn), the total rural area accounts for >98% of the domestic area.
Evaluations at rural sites (Table S4) show a mean bias of -5% (Fig. 3). The magnitude of
such bias is much lower than the values of comparisons at urban-dominant sites, where
simulated $[O_3]$ is higher by 42.5% for the summer mean (Fig. 2f) and 55.6% for the
annual mean (Fig. S5f).

### 359    3.1.3 Shortwave radiation

We use ground-based observations of surface shortwave radiation and diffuse fraction
from 106 pyranometer sites managed by the Climate Data Center, Chinese
Meteorological Administration (Xia, 2010). Site selection is based on the availability of
continuous monthly measurements during 2008-2012, resulting in 95 sites for the
evaluation of total shortwave radiation. For diffuse radiation, we select the 17 sites only
that provide continuous measurements during 2008-2012. Simulated surface shortwave
radiation agrees well with measurements at 106 sites for both summer (Figs 4a-4c) and
annual (Figs S6a-S6c) means. Simulated diffuse fraction reproduces observed spatial
pattern with high correlation coefficient ($r = 0.74$ for summer and $r = 0.65$ for annual, $p <$
0.01), though it is larger than observations on average by 25.2% in summer (Figs 4d-4f)
and 35.2% for the annual mean (Figs S6d-S6f). Such bias is mainly attributed to the
overestimation in the North and Northeast. For the southeastern region, where high
values of GPP dominate (Fig. 1), predicted diffuse fraction is in general within the 10%
difference from the observations.

**3.1.4 Ozone vegetation damage function**

We adopt the same approach as Yue et al. (2016) by comparing simulated GPP-to-[$O_3$] responses (Table S2) with observations from multiple published literature (Table S1). We aggregate these measurements into six categories, including evergreen needleleaf forest (ENF), deciduous broadleaf forest (DBF), shrubland, C3 herbs, C4 herbs, and a mixture of all above species. We derive the sensitivity of GPP to varied [$O_3$] (Fig. 5) using the YIBs offline version. For most PFTs, simulated $O_3$ damage increases with [$O_3$] in broad agreement with measurements. Predicted $O_3$ damage reproduces observations with a correlation coefficient of 0.61 (for all samplings, *n*=32) and in similar magnitudes (-17.7% vs. -20.4%), suggesting that the damage scheme we adopted from Sitch et al. (2007) is ready to use in China. For the same level of [$O_3$], deciduous trees suffer larger damages than evergreen trees because the former species are usually more sensitive (Sitch et al., 2007) and have higher $g_s$ (therefore higher uptake) (Wittig et al., 2007). Flux-based $O_3$ sensitivity for C4 herbs is only half that of C3 herbs (Sitch et al., 2007), however, concentration-based $O_3$ damages, both observed and simulated, are larger for C4 plants because of their higher uptake efficiency following high $g_s$ (Yue and Unger, 2014).

**3.2 $O_3$ effects in China**

We focus our study domain in eastern China (21°-38°N, 102°-122°E, including the North China Plain, the Yangtze River Delta, and part of the Sichuan Basin), a region that suffers from high levels of $O_3$ and aerosols from anthropogenic pollution sources (>75% contribution; Fig. S7). We estimate that surface $O_3$ decreases annual GPP in China by 10.3% based on YIBs offline simulations in the absence of feedbacks from $O_3$ vegetation damage to meteorology and plant growth. The damage is stronger in summer, when the average GPP decreases by ~20% for both deciduous trees and C3 herbs in the East (Fig. 6). In contrast, a lower average damage to GPP of ~10% is predicted for evergreen needleleaf trees (because of low sensitivity) and C4 herbs (because of the mismatched spatial locations between C4 plants and high [$O_3$], Fig. S1 and Fig. 2d).


$O_3$ damage to photosynthesis can influence plant growth. At the same time, the $O_3$-
induced reductions in stomatal conductance (Fig. S8a) can increase canopy temperature
and inhibit plant transpiration, leading to surface warming (Fig. S8b), dry air (Fig. S8c),
and rainfall deficit (Fig. S8d). These biometeorological feedbacks may in turn exacerbate
the dampening of land carbon uptake. Application of ModelE2-YIBs that allows for these
feedbacks gives an $O_3$-induced damage to annual GPP of 10.7%, a similar level to the
damage computed in YIBs offline. The spatial pattern of the online $O_3$ inhibition also
resembles that of offline damages (not shown). Sensitivity simulations with zero
anthropogenic emissions show almost no $O_3$ damage (Fig. S9), because the $[O_3]$ exposure
from natural sources alone is usually lower than the threshold level of 40 ppbv below
which the damage for most PFTs is limited (Fig. 5). Our results indicate that present-day
surface $O_3$ causes strong inhibitions on total NPP in China, ranging from $0.43 \pm 0.12$ Pg
C $yr^{-1}$ with low sensitivity to $0.76 \pm 0.15$ Pg C $yr^{-1}$ with high sensitivity (Table 3). The
central value of NPP reduction by $O_3$ is $0.59 \pm 0.11$ Pg C $yr^{-1}$, assuming no direct impacts
of $O_3$ on plant respiration. About 61% of such inhibition occurs in summer, when both
photosynthesis and $[O_3]$ reach maximum of the year.

**3.3 Aerosol haze effects in China**

Aerosols decrease direct solar radiation but increase diffuse radiation (Fig. S10), the latter
of which is beneficial for canopy photosynthesis. The online-coupled model quantifies
the concomitant meteorological and hydrological feedbacks (Fig. 7) that further influence
the radiative and land carbon fluxes. Reduced insolation decreases summer surface
temperature by 0.63°C in the East, inhibiting evaporation but increasing relative humidity
(RH) due to the lower saturation vapor pressure (Table S5). These feedbacks combine to
stimulate photosynthesis (Fig. 8a), which, in turn, strengthens plant transpiration (not
shown). Atmospheric circulation and moisture convergence are also altered due to the
pollution-vegetation-climate interactions, resulting in enhanced precipitation (Fig. 7b)
and cloud cover (Fig. 7d). Moreover, soil moisture increases (Fig. 7f) with lower
evaporation (Fig. 7e) and higher precipitation (Fig. 7b). Inclusion of AIE results in
distinct climatic feedbacks (Fig. S11). Summer precipitation decreases by 0.9 mm day$^{-1}$
(13%), leading to a 3% decline in soil moisture (Table S6). The AIE lengthens cloud
lifetime and increases cloud cover, further reducing available radiation and causing a
stronger surface cooling. Compared to aerosol-induced perturbations in radiation and
temperature, responses in hydrological variables (e.g. precipitation and soil moisture) are
usually statistically insignificant on the domain average due to the large relative
interannual climate variability (Tables S5 and S6). The resulting meteorological changes
over China are a combination of locally driven effects (such as changes in radiation and
hence temperature) and regional-globally driven effects (such as changes in rainfall and
hence soil water).

We separate the relative impacts due to aerosol-induced perturbations in temperature,
radiation, and soil moisture (Fig. 8). The overall impact of the aerosol-induced
biometeorological feedbacks on the carbon uptake depends on the season and vegetation
type. In the summer, the aerosol-induced surface cooling brings leaf temperature closer to
the photosynthetic optimum of 25°C, DRF enhances light availability of shaded leaves
and LUE of sunlit leaves, and the wetter soil alleviates water stress for stoma.
Consequently, aerosol-induced hydroclimatic feedbacks promote ecosystem NPP, albeit
with substantial spatiotemporal variability (Fig. 8a and Table 3). Surface cooling
enhances NPP in summer (Fig. 8b) but induces neutral net impacts on NPP in spring and
autumn (not shown), when leaf temperature is usually below 25°C, because the cooling-
driven reductions of photosynthesis are accompanied by simultaneous reductions in plant
respiration. We find strong aerosol DRF in the Southeast and the Northeast, where AOD
is moderate (Fig. 8c). Over the North China Plain and the Southwest, aerosol DRF is
more limited. In these regions, the local background aerosol layer and/or cloud over are
sufficiently optically thick that the effect of anthropogenic aerosol pollution is largely to
attenuate direct sunlight and reduce NPP (Cohan et al., 2002). Aerosol-induced cooling
increases soil moisture over most of the East (Fig. 7f), but the beneficial responses are
confined to the Central East (Fig. 8d), where C3 crops dominate (Fig. S1). These short-
root plants are more sensitive to short-term water availability than deep-root trees (Beer
et al., 2010; Yue et al., 2015).

In contrast, inclusion of AIE results in detrimental impacts on NPP (Table 3). Aerosol-
induced drought strongly reduces regional NPP especially over the Northeast and North
China Plain (Fig. S12d), where cropland dominates (Fig. S1). Meanwhile, the increases
in cloud cover reduce available radiation, leading to weakened aerosol DRF effects over
the Southeast while strengthened NPP reductions in the Southwest (Fig. S12c).

**3.4 Combined effects of $O_3$ and aerosol**

Simultaneous inclusion of the aerosol effects on the land biosphere has negligible impacts
on $O_3$ damage. The online $O_3$ inhibition, which is much stronger in magnitude than the
aerosol effects, shows insignificant differences relative to the offline values (10.7% vs.
10.3%). As a result, we consider $O_3$ and aerosol effects to be linearly additive. In the year
2010, the combined effects of $O_3$ and aerosols (Table 3) decrease total NPP in China by
0.39 (without AIE) to 0.80 Pg C yr$^{-1}$ (with AIE), equivalent to 9-16% of the pollution-
free NPP and 16-32% of the total anthropogenic carbon emissions (Liu et al., 2015).
Spatially, a dominant fraction (86% without AIE and 77% with AIE) of the reduced
carbon uptake occurs in the East, where dense air pollution is co-located with high NPP
(Figs 1 and 2). Temporally, a dominant fraction (60% without AIE and 52% with AIE) of
the reduced carbon uptake occurs in summer, when both NPP and $[O_3]$ reach maximum
of the year. Independently, $O_3$ reduces NPP by 0.59 Pg C yr$^{-1}$, with a large range from
0.43 Pg C yr$^{-1}$ for low damaging sensitivity to 0.76 Pg C yr$^{-1}$ for high damaging
sensitivity (Table 3). The sign of the aerosol effects is uncertain. Without AIE, aerosol is
predicted to increase NPP by 0.2 Pg C yr$^{-1}$, because of regionally confined DRF effects
and enhanced soil moisture (Fig. 8). With inclusion of AIE, aerosol decreases NPP by 0.2
Pg C yr$^{-1}$, mainly due to reduced soil moisture (Fig. S12). The uncertainty of individual
simulations, calculated from the interannual climate variability, is usually smaller than
that due to $O_3$ damage sensitivity and AIE (Table 3).

**3.5 Future projection of pollution effects**

Following the CLE scenario, by the year 2030, predicted summer $[O_3]$ increases by 7%,
while AOD decreases by 5% and surface $PM_{2.5}$ concentrations decline by 10% (Fig. 9).
These changes are predominantly attributed to changes in anthropogenic emissions, as
natural sources show limited changes. The reduction of AOD is related to the decreased
emissions of $SO_2$, black carbon, and organic carbon (Fig. S2). In contrast, the
enhancement of $[O_3]$ results from the increased $NO_x$ emissions, higher level of
background $CH_4$ (~20%), and higher air temperature in the warmer 2030 climate. The
moderate decline of aerosol loading in the 2030 CLE scenario brings benefits to land
ecosystems through DRF effects (Table 3) because light scattering is often saturated in
the present-day conditions due to high local AOD and regional cloud cover. Benefits
from the aerosol pollution reductions are offset by worsening $O_3$ vegetation damage in
the CLE future world (Fig. 10b). $O_3$-free ($[O_3]$=0) NPP increases by 14% in 2030 due to
$CO_2$ fertilization and global climate change. Despite $[CO_2]$ increases from 390 ppm in
2010 to 449 ppm in 2030 in the RCP8.5 scenario (van Vuuren et al., 2011), which
contributes to $g_s$ inhibition of 4% on the country level, the future $O_3$-induced NPP
damage in 2030 degrades to 14% or 0.67 Pg C $yr^{-1}$ (Table 3), with a range from 0.43 Pg
C $yr^{-1}$ (low $O_3$ sensitivity) to 0.90 Pg C $yr^{-1}$ (high $O_3$ sensitivity).

The MTFR scenario reflects an ambitious and optimistic future in which there is rapid
global implementation of all currently available technological pollution controls. AOD
decreases by 55% and $[O_3]$ decreases by 40% for this future scenario (Fig. 9). The model
projects much lower damage to NPP of only 0.12 Pg C $yr^{-1}$, with a range from 0.06 Pg C
$yr^{-1}$ (low $O_3$ sensitivity) to 0.20 Pg C $yr^{-1}$ (high $O_3$ sensitivity), mainly due to the 40%
reduction in $[O_3]$ (Fig. 10c). Including both aerosol direct and indirect effects, $O_3$ and
aerosols together inhibit future NPP by 0.28 Pg C $yr^{-1}$, ranging from 0.12 Pg C $yr^{-1}$ with
low $O_3$ sensitivity to 0.43 Pg C $yr^{-1}$ with high $O_3$ sensitivity. As a result, The MTFR
scenario offers strong recovery of the land carbon uptake in China by 2030.

**4. Discussion**

**4.1 Comparison with previous estimates**

Previous estimates of $O_3$ damages over the whole China region are very limited. Two
important studies, Tian et al. (2011) and Ren et al. (2011), have quantified the impacts of
surface $O_3$ on carbon assimilation in China. Both studies applied the dynamic land
ecosystem model (DLEM) with $O_3$ damage scheme proposed by Felzer et al. (2004),
except that Tian et al. (2011) focused on NEE while Ren et al. (2011) also investigated
NPP. The Felzer et al. (2004) scheme calculates $O_3$ uptake based on stomatal
conductance and the AOT40 (accumulated hourly $O_3$ dose over a threshold of 40 ppb).
Yue and Unger (2014) estimated $O_3$-induced reductions in GPP over U.S. using Sitch et
al. (2007) scheme and found an average value of 4-8% (low to high sensitivity),
consistent with the reduction of 3-7% in Felzer et al. (2004). For this study, we estimate
that present-day $O_3$ decreases NPP by 0.43-0.76 Pg C yr$^{-1}$, higher than the 0.42 Pg C yr$^{-1}$
calculated by Ren et al. (2011). However, the percentage reduction of 10.1-17.8% in our
estimate is weaker than the value of 24.7% in Ren et al. (2011). The main reason for such
discrepancy lies in the differences in the climatological NPP. Combining all
environmental drivers (e.g. $[CO_2]$, meteorology, $[O_3]$, and aerosols), we predict an
average NPP of $3.98 \pm 0.1$ Pg C yr$^{-1}$ for the year 2010 (uncertainties from AIE) with the
ModelE2-YIBs model. This value is close to the average of $3.35 \pm 1.25$ Pg C yr$^{-1}$ for
1981-2000 calculated based on 54 estimates from 33 studies (Shao et al., 2016). Using
DLEM, Ren et al. (2011) estimated an optimal NPP of 1.67 Pg C yr$^{-1}$ for 2000-2005 over
China, which is only half of the literature-based estimate.

In the absence of any previous studies of aerosol pollution effects on land carbon uptake
in China, our strategy is to compare separately the simulated aerosol climatic feedback
(climate sensitivity) and simulated NPP response to climate variability (NPP sensitivity)
with existing published results. ModelE2-YIBs simulates an annual reduction of 26.2 W
m$^{-2}$ in all-sky surface solar radiation over the East due to aerosols pollution (Table S5),
similar to the estimate of 28 W m$^{-2}$ by Folini and Wild (2015). In response to this
radiative perturbation, aerosol pollution causes a widespread cooling of 0.3-0.9 °C in
summer over the East (Fig. 7a), consistent with estimates of 0-0.9 °C by Qian et al.
(2003), 0-0.7 °C by  Liu et al. (2009), and average of 0.5 °C by Folini and Wild (2015).
Aerosol pollution effects on regional precipitation patterns in China are not well
understood due to different climate model treatments of land-atmosphere interactions and
the interplay between regional and large-scale circulation. In ModelE2-YIBs, without
AIE, aerosol induces a "northern drought and southern flood" pattern in agreement with
Gu et al. (2006), but different to Liu et al. (2009) who predicted widespread drought
instead. Including both aerosol direct and indirect effects, ModelE2-YIBs simulates an
average reduction of 0.48 mm day$^{-1}$ in summer rainfall widespread over China (Fig.
S11b), similar to the magnitude of 0.4 mm day$^{-1}$ estimated with the ECHAM5-HAM
model (Folini and Wild, 2015), but higher than the 0.21 mm day$^{-1}$ predicted by the
RegCM2 model (Huang et al., 2007).

Sensitivity experiments with YIBs show that summer NPP increases following aerosol-
induced changes in temperature, radiation, and precipitation (Fig. 8). The cooling-related
NPP enhancement (Fig. 8b) collocates with changes in temperature (Fig. 7a), indicating
that sensitivity of NPP to temperature is negative over eastern China. Such temperature
sensitivity is consistent with the ensemble estimate based on 10 terrestrial models (Piao et
al., 2013). For the aerosol-induced radiative perturbation, many studies have shown that
moderate aerosol/cloud amount promotes plant photosynthesis through enhanced DRF,
while dense aerosol/cloud decreases carbon uptake due to light extinction (Cohan et al.,
2002; Gu et al., 2003; Rocha et al., 2004; Alton, 2008; Knohl and Baldocchi, 2008;
Mercado et al., 2009; Jing et al., 2010; Bai et al., 2012; Cirino et al., 2014; Strada et al.,
2015). Theoretically, at each specific land location on the Earth, there is an AOD
threshold below which aerosol promotes local NPP. The threshold is dependent on
latitude, cloud/aerosol amount, and plant types. In a related study by Yue and Unger
(2017), we applied a well-validated offline radiation model to calculate these AOD
thresholds over China. We conclude that present-day AOD is lower than the local
thresholds in the Northeast and Southeast but exceeds the thresholds in the North China
Plain, explaining why aerosol-induced diming enhances NPP in the former regions but
reduces NPP in the latter (Fig. 8c). On the country level, the NPP enhancement due to
aerosol DRF is 0.07 Pg C yr$^{-1}$ in Yue and Unger (2017), very close to the 0.09 Pg C yr$^{-1}$
estimated with ModelE2-YIBs model (Table 2).

**4.2 Uncertainties**


A major source of uncertainty originates from the paucity of observations. For instance,
direct measurements of aerosol pollution effects on NPP are non-existent for China. The
aerosol effects involve complex interactions that challenge the field-based validation of
the underlying independent processes. Field experiments of $O_3$ vegetation damage are
becoming more available, but their applications are limited by the large variations in the
species-specific responses (Lombardozzi et al., 2013), as well as the discrepancies in the
treatments of [$O_3$] enhancement (Wittig et al., 2007). Instead of equally using all
individual records from multiple literatures (Lombardozzi et al., 2013), we aggregate $O_3$
damage for each literature based on the seasonal (or growth-season) average. In this way,
the derived PFT-level GPP-[$O_3$] relationships are not biased towards the experiments
with a large number of samplings. Such aggregation also reduces sampling noise and
allows construction of the quantified GPP-[$O_3$] relationships used for model assessment.
Predicted [$O_3$] is largely overestimated at urban sites but exhibits reasonable magnitude at
rural sites (Figs 2 and 3). Measurements of background [$O_3$] in China are limited both in
space and time, restricting comprehensive validation of [$O_3$] and the consequent estimate
of $O_3$ damages on the country level.

We have estimated $O_3$ damages to NPP (instead of GPP), an optimal indicator for net
carbon uptake by plants. Our calculations assume no impacts of $O_3$ on autotrophic
respiration. Yet, limited observations have found increased plant respiration in response
to $O_3$ injury (Felzer et al., 2007), suggesting that our calculation of $O_3$-induced NPP
reductions might be underestimated. Current large mechanistic uncertainties in the role of
anthropogenic nitrogen (N) deposition to China's land carbon uptake (Tian et al., 2011;
Xiao et al., 2015) have prohibited the inclusion of dynamic carbon-nitrogen coupling in
the Earth system model used here. Previous studies have suggested that inclusion of N
fertilization can relieve or offset damages by $O_3$, especially for N-limited forests
(Ollinger et al., 2002). Relative to the present day, atmospheric reactive N deposition
increases by 20% in the CLE scenario but decreases by 60% in the MTFR scenario,
suggesting that the stronger $O_3$ damage in CLE might be overestimated while the reduced
damage in MTFR might be too optimistic.

Our estimate of NPP responses to aerosol pollution is sensitive to modeling uncertainties
in concentration, radiation, and climatic effects. Simulated surface $PM_{2.5}$ is reasonable but
AOD is underestimated in the North China Plain (Fig. 2), likely because of the biases in
aerosol optical parameters. Using a different set of optical parameters, we predicted much
higher AOD that is closer to observations with the same aerosol vertical profile and
particle compositions (Yue and Unger, 2017). The model overestimates diffuse fraction
in China (Fig. 4), likely because of simulated biases in clouds. Previously, we improved
the prediction of diffuse fraction in China using observed cloud profiles for the region
(Yue and Unger, 2017). Biases in simulated AOD and diffuse fraction introduce
uncertainties in the aerosol DRF especially in the affected localized model grid cells. Yet,
averaged over the China domain, our estimate of NPP change by aerosol DRF (0.09 Pg C
$yr^{-1}$) is consistent with the previous assessment in Yue and Unger (2017) (0.07 Pg C $yr^{-1}$).
Aerosol-induced impacts on precipitation and soil moisture are not statistically significant
over the regionally averaged domain (Tables S5 and S6). However, for the 2010 and
2030 CLE cases with AIE, 2 out of 6 scenarios, the aerosol-induced impact on soil
moisture dominates the total NPP response (Table 3). Furthermore, the relatively coarse
resolution of the global model and usage of emission inventories may introduce
additional biases and exacerbate the total uncertainties.

Importantly, our estimate of NPP response to aerosol effects, with or without AIE, is
secondary in magnitude compared to the $O_3$ vegetation damage, suggesting that the net
impact of current air pollution levels in China is detrimental to the land carbon uptake
there. Locally, this pollution damage exerts a threat to plant health, terrestrial ecosystem
services, and food production. Globally, air pollution effects may enhance planetary
warming by decreasing the land removal rate of atmospheric $CO_2$. Our results show
substantial benefits to the protection of plant health and the regional land carbon sink in
China from stringent air pollution controls, especially for $O_3$ precursors. Our analysis
highlights the complex interplay between immediate and more local pollution issues, and
longer-term global warming. Future air pollution controls provide an additional co-
benefit to human society: the offsetting of fossil fuel $CO_2$ emissions through enhanced
land sequestration of atmospheric $CO_2$.

*Acknowledgements.* The authors are grateful to Prof. William Collins and an anonymous
reviewer for constructive comments improving this paper. X. Yue acknowledges funding
support from the National Basic Research Program of China (973 program, Grant No.
2014CB441202) and the "Thousand Youth Talents Plan". This research was supported in
part by the facilities and staff of the Yale University Faculty of Arts and Sciences High
Performance Computing Center.

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

**Table 1.** Summary of models and simulations

| Model Name | Model class | Climate drivers | Number of runs | Table index [a] | Purpose |
|---|---|---|---|---|---|
| ModelE2-YIBs | Coupled climate model | Online | 24 | 2 | Calculate $\Delta$NPP by $O_3$ and aerosols at 2010 and 2030 |
| YIBs | Vegetation model | MERRA | 15 | S2 | Evaluate $O_3$ damage scheme for China PFTs |
| YIBs | Vegetation model | ModelE2-YIBs | 30 | S3 | Isolate aerosol individual climatic impacts on NPP |

[a] Table index refers to the tables in the main text and supporting information.


**Table 2.** Summary of 24 online simulations with the ModelE2-YIBs model

| Simulations | Period | Emission Inventories | Emission sources | Ozone damage | Aerosol indirect effect |
|---|---|---|---|---|---|
| G10NATNO3 | 2010 | GAINS [a] | Natural | Null | No |
| G10ALLNO3 | 2010 | GAINS | All [d] | Null | No |
| G10ALLLO3 | 2010 | GAINS | All | Low | No |
| G10ALLHO3 | 2010 | GAINS | All | High | No |
| G30NATNO3 | 2030 | GAINS CLE [b] | Natural | Null | No |
| G30ALLNO3 | 2030 | GAINS CLE | All | Null | No |
| G30ALLLO3 | 2030 | GAINS CLE | All | Low | No |
| G30ALLHO3 | 2030 | GAINS CLE | All | High | No |
| M30NATNO3 | 2030 | GAINS MTFR [c] | Natural | Null | No |
| M30ALLNO3 | 2030 | GAINS MTFR | All | Null | No |
| M30ALLLO3 | 2030 | GAINS MTFR | All | Low | No |
| M30ALLHO3 | 2030 | GAINS MTFR | All | High | No |
| G10NATNO3_AIE | 2010 | GAINS | Natural | Null | Yes |
| G10ALLNO3_AIE | 2010 | GAINS | All | Null | Yes |
| G10ALLLO3_AIE | 2010 | GAINS | All | Low | Yes |
| G10ALLHO3_AIE | 2010 | GAINS | All | High | Yes |
| G30NATNO3_AIE | 2030 | GAINS CLE | Natural | Null | Yes |
| G30ALLNO3_AIE | 2030 | GAINS CLE | All | Null | Yes |
| G30ALLLO3_AIE | 2030 | GAINS CLE | All | Low | Yes |
| G30ALLHO3_AIE | 2030 | GAINS CLE | All | High | Yes |
| M30NATNO3_AIE | 2030 | GAINS MTFR | Natural | Null | Yes |
| M30ALLNO3_AIE | 2030 | GAINS MTFR | All | Null | Yes |
| M30ALLLO3_AIE | 2030 | GAINS MTFR | All | Low | Yes |
| M30ALLHO3_AIE | 2030 | GAINS MTFR | All | High | Yes |

[a] GAINS is short for the v4a emission inventory of Greenhouse Gas and Air Pollution
Interactions and Synergies (http://gains.iiasa.ac.at/models/index.html).
[b] CLE is the emission scenario predicted based on current legislation emissions.
[c] MTFR is the emission scenario predicted with maximum technically feasible
reductions.
[d] All emissions including both natural and anthropogenic sources. For the detailed
anthropogenic emissions, refer to Fig. S2.

**Table 3.** Changes in NPP over China due to combined and separate effects [a] of air
pollution  (units: Pg C yr$^{-1}$)

|  | 2010 | 2030 CLE | 2030 MTFR |
|---|---|---|---|
| **O$_3$ (mean)** [b] | **-0.59 ± 0.11 (-0.60 ± 0.13)** | **-0.67 ± 0.11 (-0.71 ± 0.16)** | **-0.29 ± 0.14 (-0.31 ± 0.10)** |
| Low sensitivity | -0.43 ± 0.12 (-0.40 ± 0.13) | -0.43 ± 0.14 (-0.51 ± 0.16) | -0.22 ± 0.17 (-0.15 ± 0.10) |
| High sensitivity | -0.76 ± 0.15 (-0.80 ± 0.16) | -0.90 ± 0.13 (-0.92 ± 0.18) | -0.36 ± 0.16 (-0.46 ± 0.12) |
| **Aerosol (total)** [c] | **0.20 ± 0.08 (-0.20 ± 0.09)** | **0.23 ± 0.14 (-0.09 ± 0.19)** | **0.16 ± 0.14 (0.04 ± 0.17)** |
| Temperature [d] | 0.03 ± 0.04 (0.01 ± 0.04) | 0.04 ± 0.02 (0.02 ± 0.05) | 0.03 ± 0.04 (0.00 ± 0.04) |
| Radiation [d] | 0.09 ± 0.04 (-0.03 ± 0.04) | 0.16 ± 0.06 (-0.01 ± 0.06) | 0.11 ± 0.04 (-0.03 ± 0.03) |
| Soil moisture [d] | 0.07 ± 0.07 (-0.19 ± 0.10) | 0.01 ± 0.09 (-0.09 ± 0.15) | 0.03 ± 0.12 (0.00 ± 0.09) |
| **O$_3$ + aerosol (net)** [e] | **-0.39 ± 0.12 (-0.80 ± 0.11)** | **-0.43 ± 0.12 (-0.80 ± 0.10)** | **-0.12 ± 0.13 (-0.28 ± 0.14)** |

[a] Results shown are the averages ± one standard deviation. Simulations with both aerosol
direct and indirect radiative effects (AIE) are shown in the brackets.
[b] Mean O$_3$ damages are calculated as half of differences in ΔNPP between low and high
sensitivities, e.g., present-day mean O$_3$ damage is $\frac{1}{2}$(G10ALLHO3+G10ALLLO3) −
G10ALLNO3.
[c] Combined aerosol effects are calculated with the ModelE2-YIBs climate model, e.g.,
present-day aerosol effect is G10ALLNO3 – G10NATNO3.
[d] Separate aerosol effects are calculated with the offline YIBs vegetation model driven
with forcings from the climate model (Table S3).
[e] The net impact of O$_3$ damages and aerosol effects, for example at present day, is
calculated as $\frac{1}{2}$(G10ALLHO3+G10ALLLO3) – G10NATNO3.

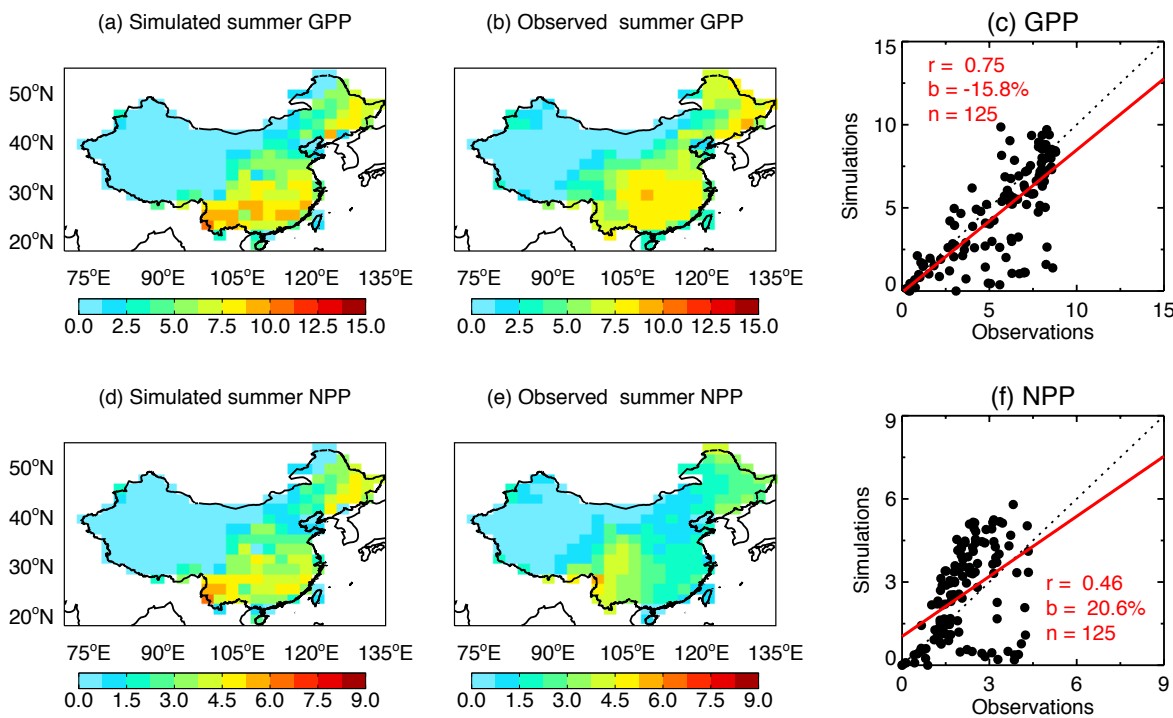

**Figure 1.** Evaluation of simulated summertime carbon fluxes by ModelE2-YIBs. Panels
show GPP (top row) and NPP (bottom row) over China. Simulation results (a, d) are the
average of G10ALLHO3 and G10ALLLO3, which are performed with the climate model
ModelE2-YIBs using high and low ozone damage sensitivities (Table 2). The correlation
coefficients (r), relative biases (b), and number of grid cells (n) for the comparisons are
listed on the scatter plots. Units: g C m$^{-2}$ day$^{-1}$.


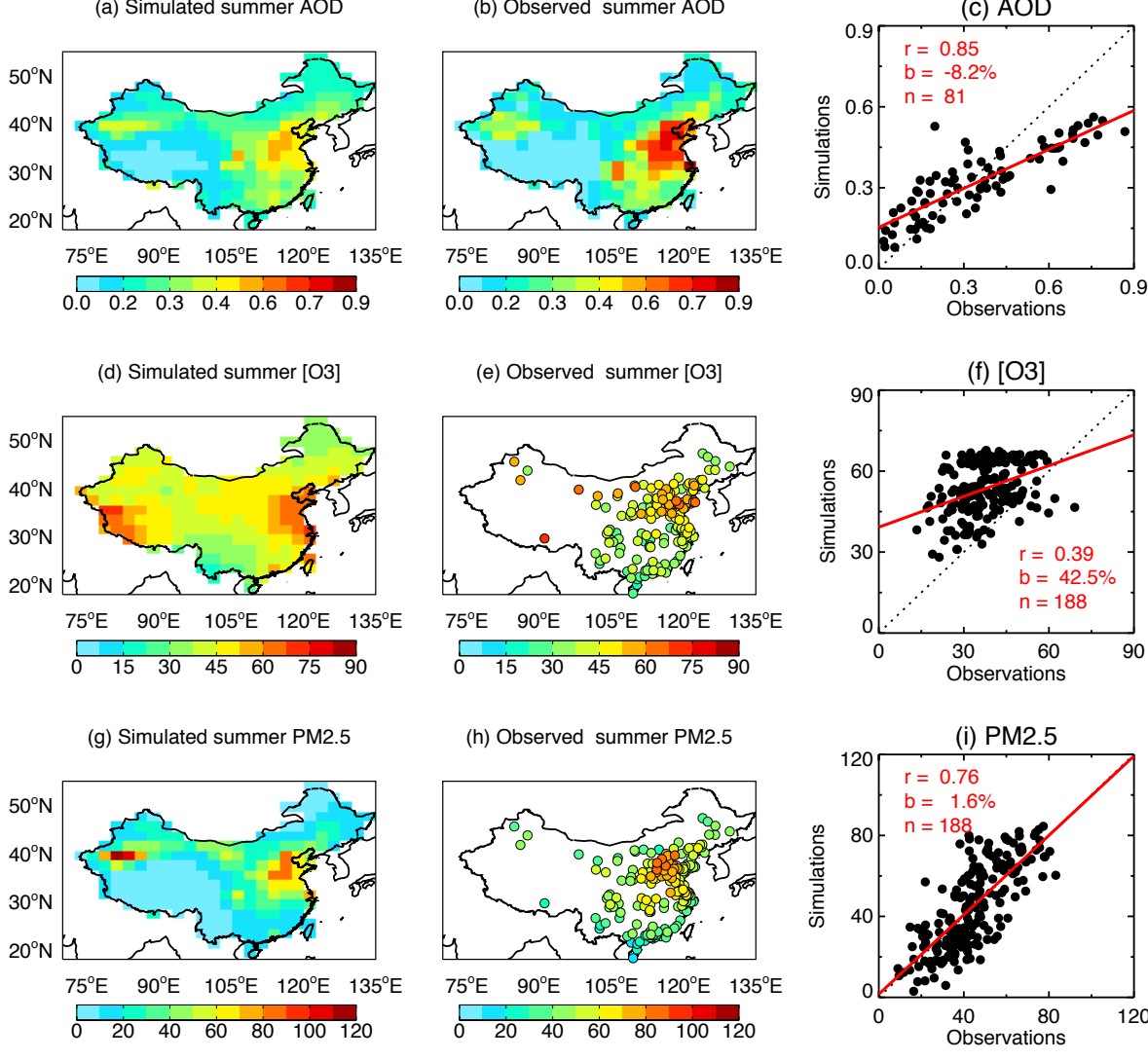


**Figure 2.** Evaluation of simulated summertime air pollution in China. Evaluations shown
include (a) aerosol optical depth (AOD) at 550 nm, (d) [O₃] (units: ppbv), and (g) PM$_{2.5}$
concentrations (units: μg m$^{-3}$) with observations from (b) the satellite retrieval of the
MODIS (averaged for 2008-2012), and (e) and (h) measurements from 188 ground-based
sites (at the year 2014). Simulation results are from G10ALLNO3 performed with the
climate model ModelE2-YIBs (Table 2). The correlation coefficients (r), relative biases
(b), and number of sites/grids (n) for the comparisons are listed on the scatter plots.



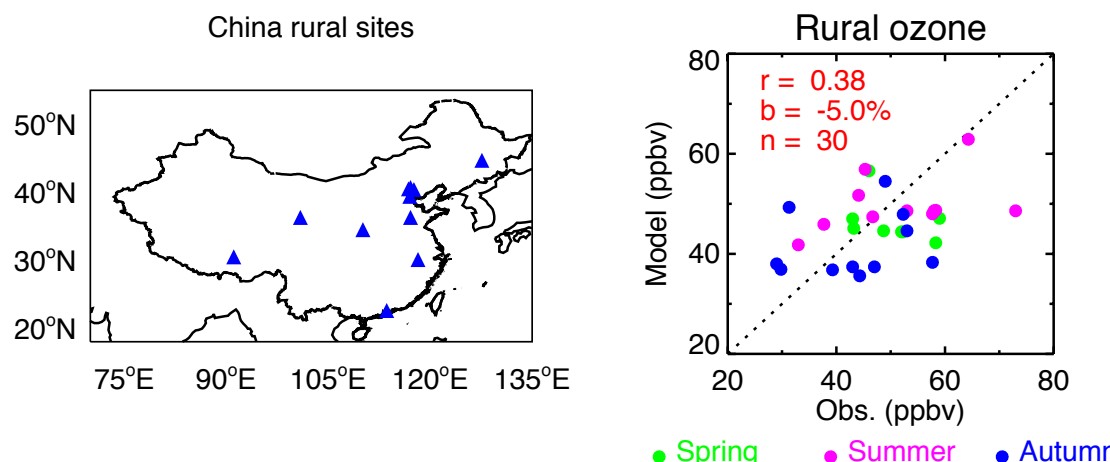

**Figure 3.** Evaluation of simulated [$O_3$] at rural sites in China. Simulation results are from
G10ALLNO3 performed with the climate model ModelE2-YIBs (Table 2). For the
scatter plots, green, pink, and blue points represent values in spring, summer, and
autumn, respectively. The data sources of all sites are listed in Table S4.


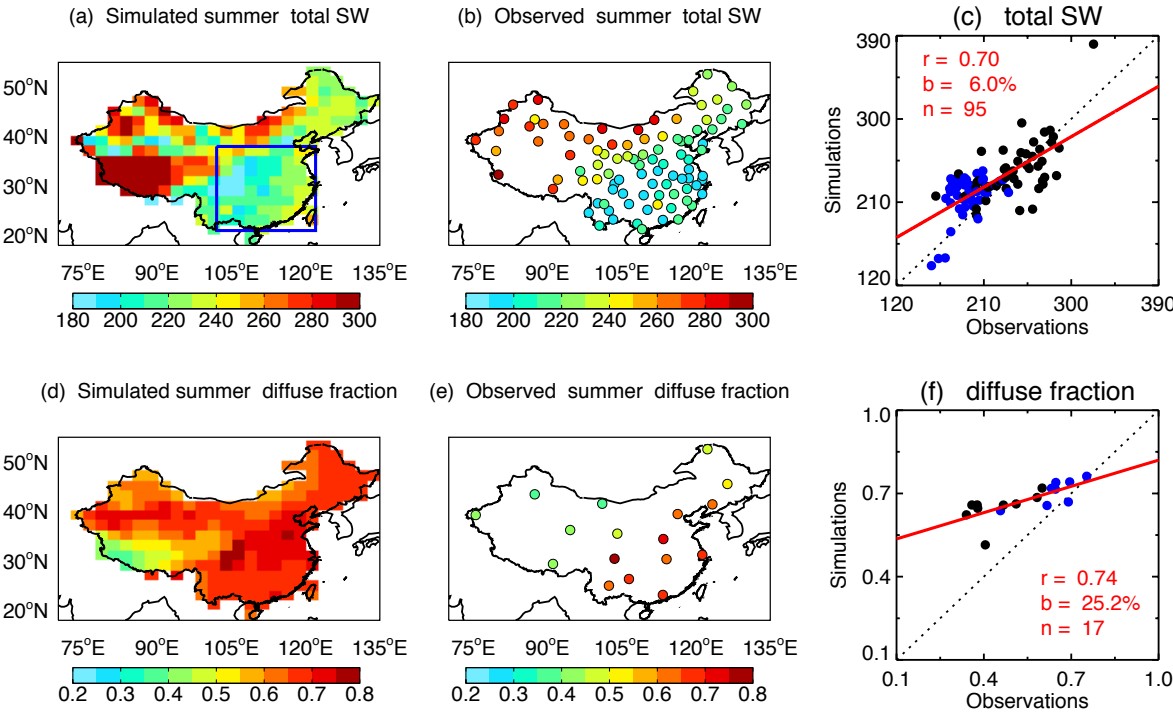

**Figure 4.** Evaluation of simulated radiation fluxes by ModelE2-YIBs. Panels show summertime surface (a) total shortwave radiation (units: W m$^{-2}$) and (d) diffuse-to-total fraction with (b, e) observations from 106 sites. Simulation results are from G10ALLNO3 performed with the climate model ModelE2-YIBs (Table 2). The correlation coefficients (r), relative biases (b), and number of sites (n) for the comparisons are listed on the (c, f) scatter plots. The blue points in the scatter plots represent sites located within the box regions in southeastern China as shown in (a).


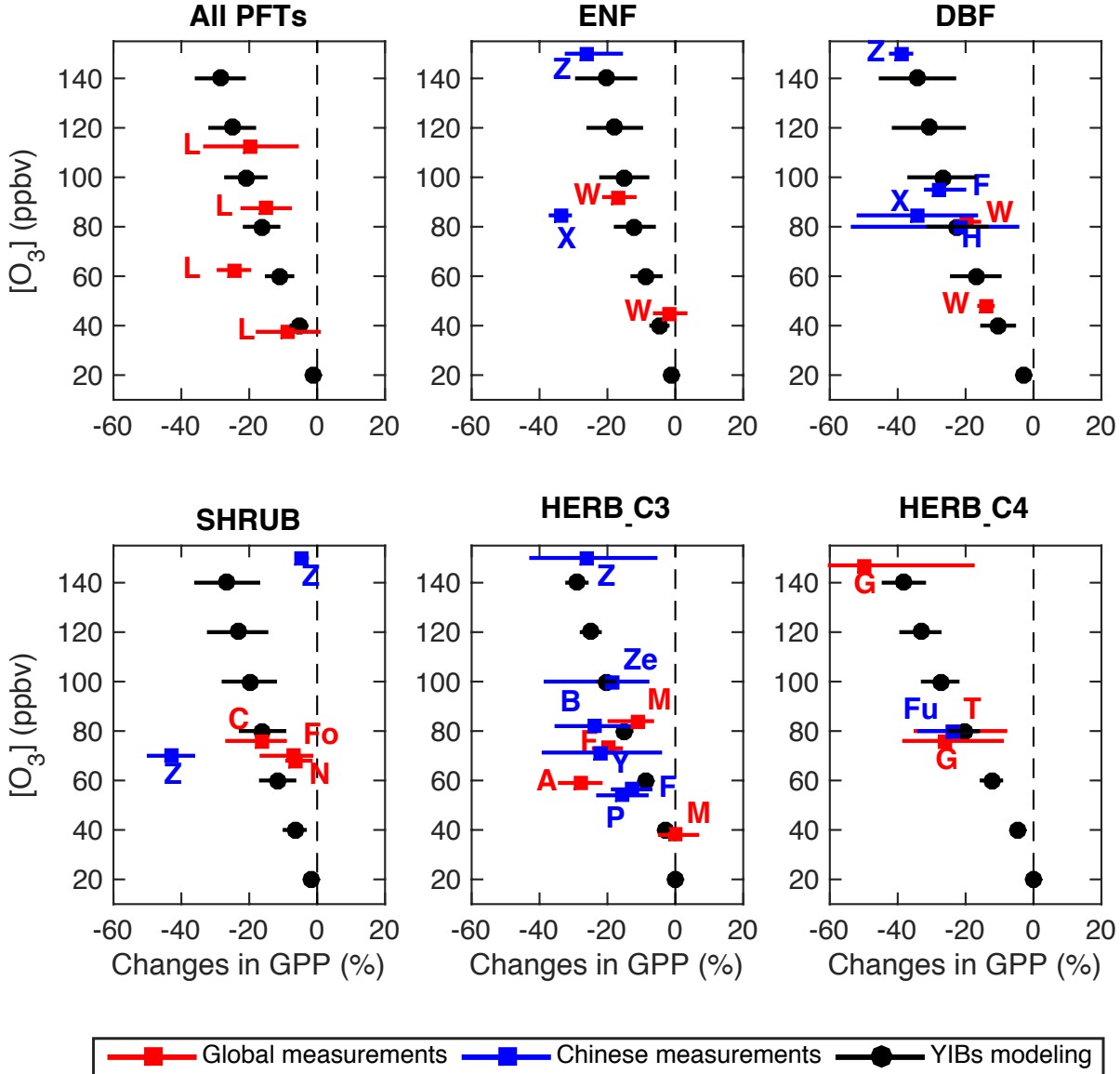

**Figure 5.** Comparison of predicted changes in summer GPP by $O_3$ with measurements. Simulations are performed using the offline YIBs vegetation model (Table S2) and averaged for all grid squares over China weighted by the area of a specific PFT. Black points show the simulated mean reductions with error bars indicating damage range from low to high $O_3$ sensitivity. Solid squares with error bars show the results (mean plus uncertainty) based on measurements reported in the literature (Table S1). Experiments performed for vegetation types in China are denoted with blue symbols. The author initials are indicated for the corresponding studies.

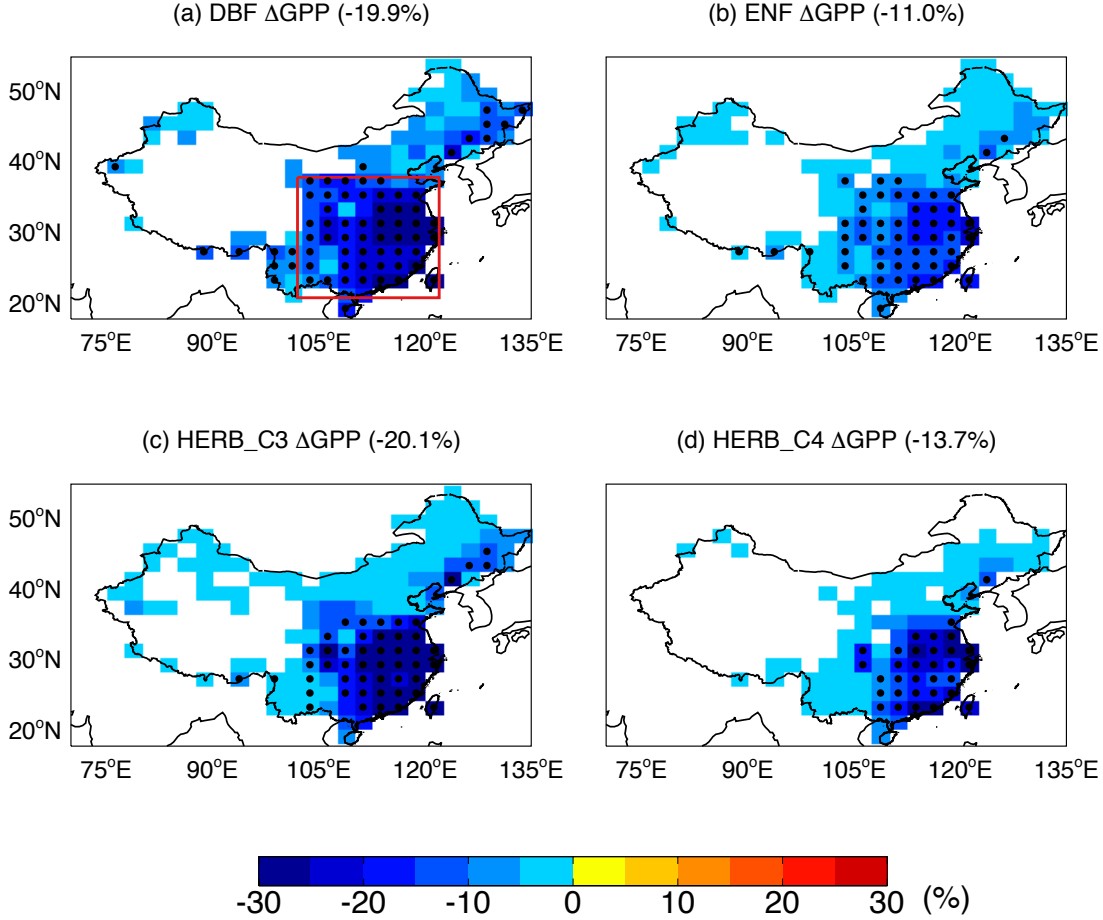

**Figure 6.** Predicted offline percentage damage to summer GPP caused by O$_3$. Panels show the damages to (a) ENF (evergreen needleleaf forest), (b) DBF (deciduous broadleaf forest), (c) C3 herbs, and (d) C4 herbs over China in the year 2010. Simulations are performed with the climate model ModelE2-YIBs, which does not feed O$_3$ vegetation damages back to affect biometeorology, plant growth, and ecosystem physiology. The results are averaged for the low and high damaging sensitivities:

$$(\tfrac{1}{2}(\text{G10ALLHO3\_OFF}+\text{G10ALLLO3\_OFF})/\text{G10ALLNO3} - 1)\times100\%$$

The average value over the box domain of (a) is shown in the title bracket of each subpanel. Significant changes ($p<0.05$) are marked with black dots.

1034

1035

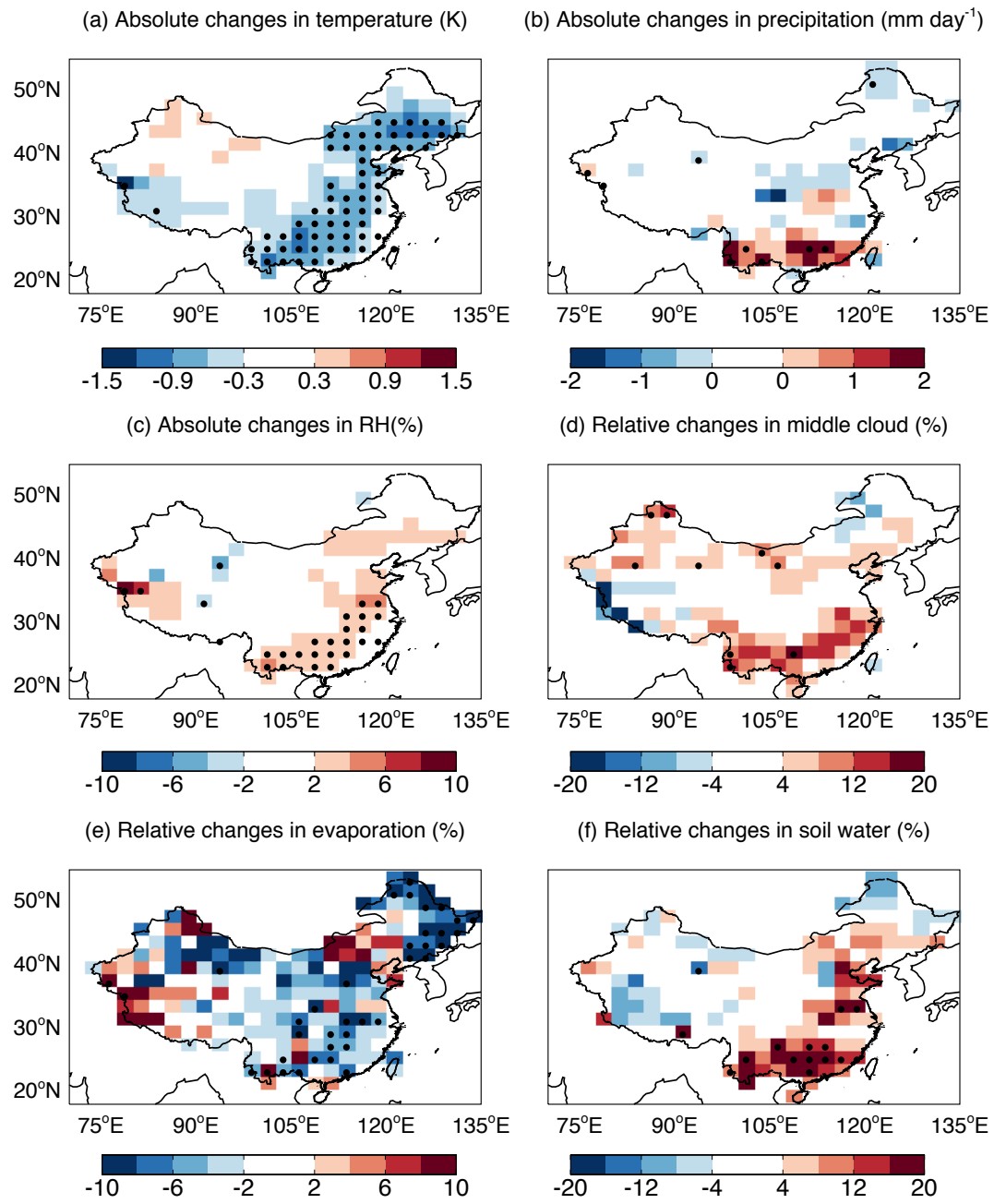

**Figure 7.** Changes in summer meteorology due to direct radiative effects of anthropogenic aerosols. All changes are calculated as the differences between the simulations G10ALLNO3 and G10NATNO3. For (a) temperature, (b) precipitation, and (c) relative humidity, we show the absolute changes as G10ALLNO3 – G10NATNO3. For (d) middle cloud cover, (e) evaporation, and (f) soil water content, we show the relative changes as (G10ALLNO3/G10NATNO3 – 1) × 100%. Significant changes ($p<0.05$) are marked with black dots.


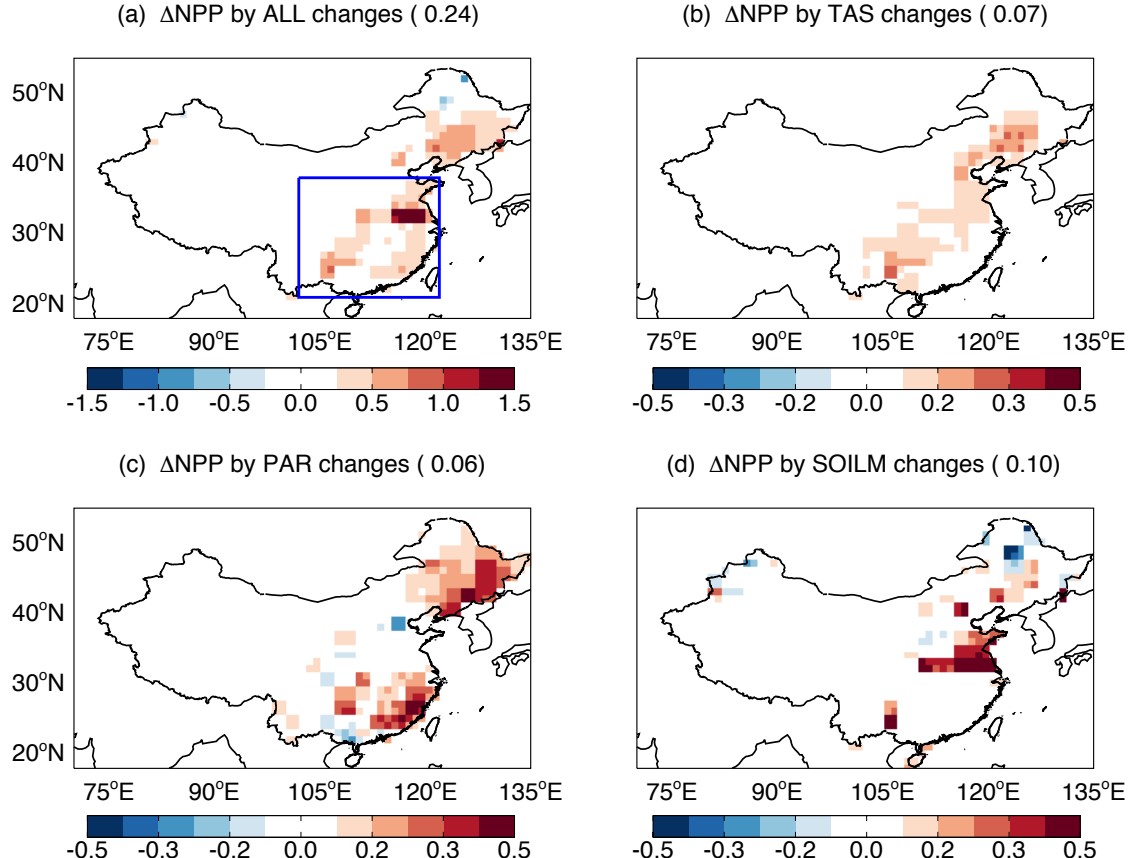

**Figure 8.** Decomposition of aerosol-induced changes in summer NPP. Changes in NPP
are caused by aerosol-induced changes in (b) surface air temperature, (c)
photosynthetically active radiation (PAR), (d) soil moisture, and (a) the combination of
above three effects. Simulations are performed with the offline YIBs vegetation model
driven with meteorological forcings simulated with the ModelE2-YIBs climate model
(Table S3). The NPP responses to PAR include the DRF effects. The color scale for the
first panel is different from the others. The average NPP perturbation over the box
domain in a is shown in the bracket of each title. Only the significant changes ($p < 0.05$)
are shown. Units: g C m$^{-2}$ day$^{-1}$.

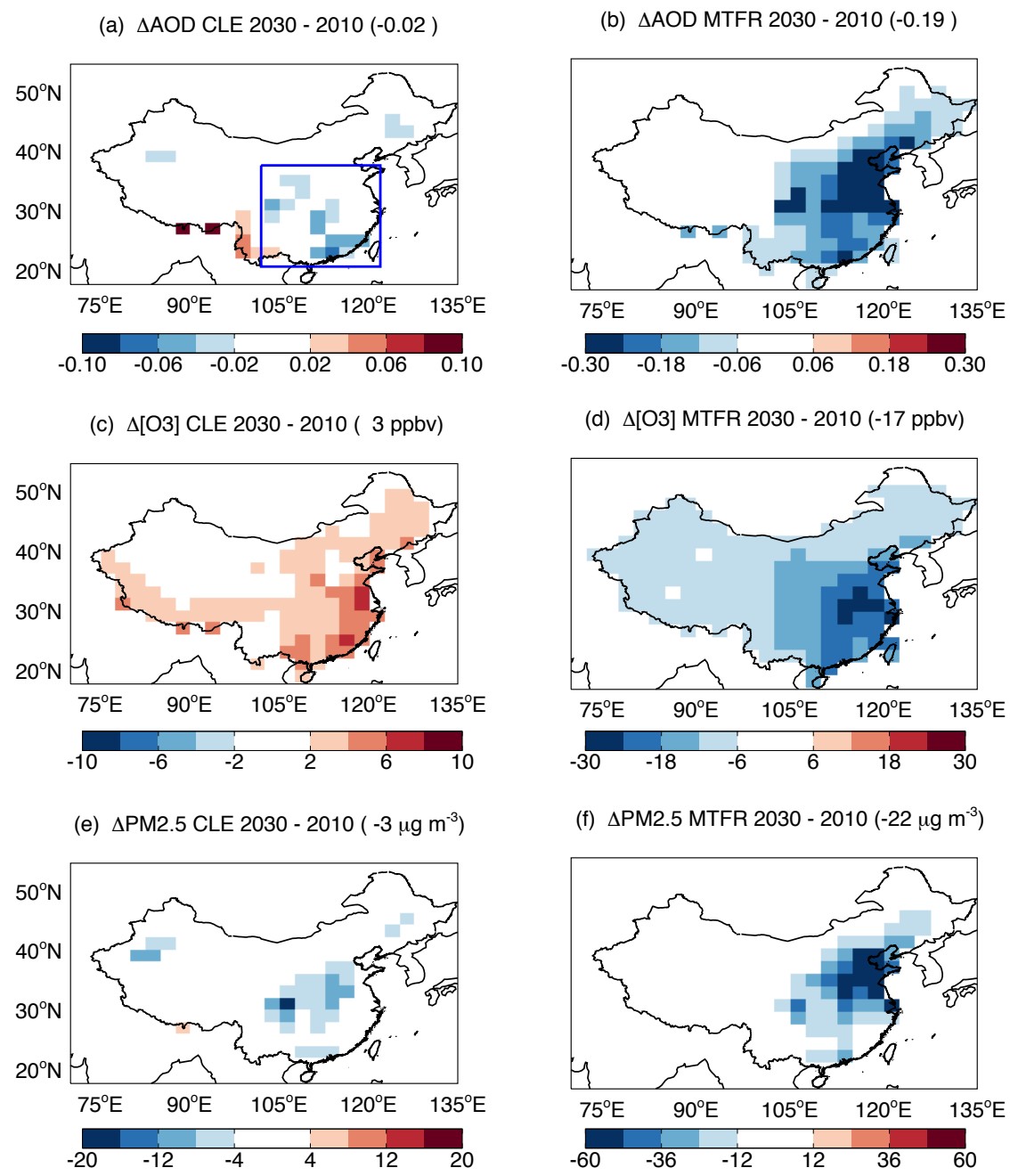

**Figure 9.** Predicted changes in summertime air pollution by 2030. Panels shown are for (a, b) AOD, (c, d) [$O_3$], and (e, f) $PM_{2.5}$ concentrations for the year 2030 relative to 2010 based on scenarios of (left) current legislation emissions (CLE) and (right) maximum technically feasible reduction (MTFR). Results for the left panels are calculated as (G30ALLNO3 – G10ALLNO3). Results for the right panels are calculated as (M30ALLNO3 – G10ALLNO3). The average value over the box domain of (a) is shown in the title bracket of each subpanel. Only the significant changes ($p < 0.05$) are shown.


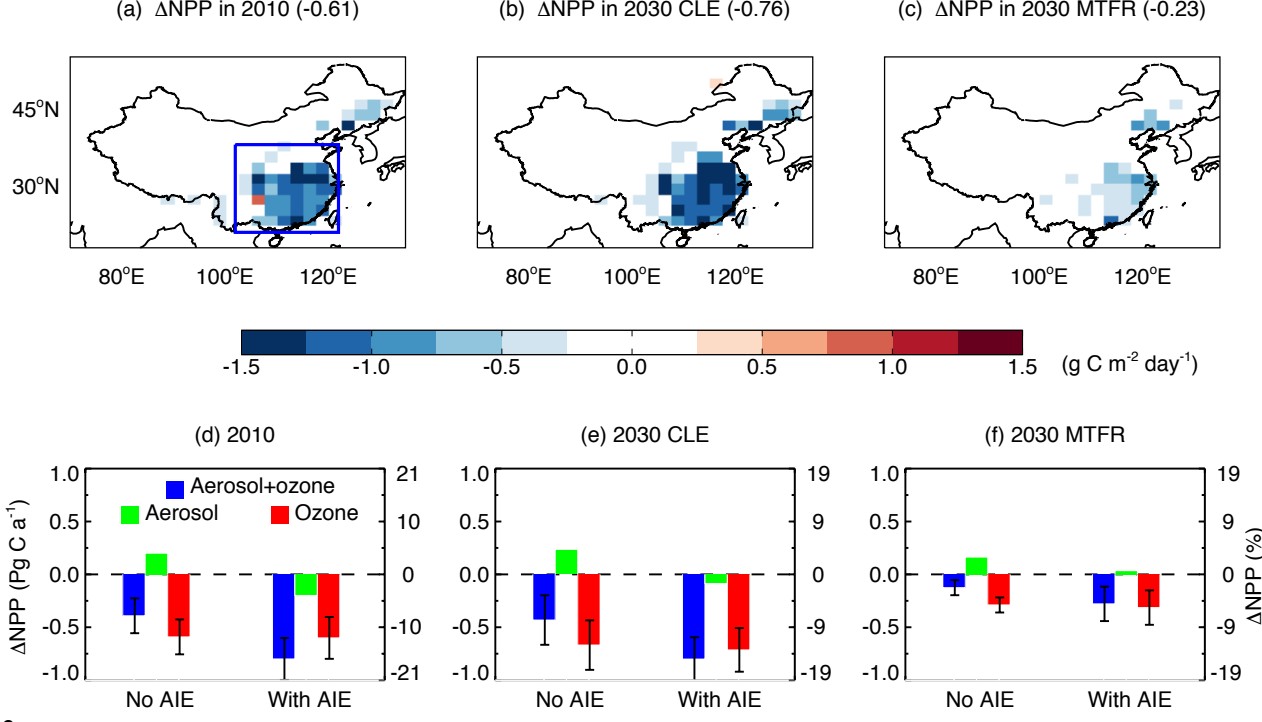


**Figure 10.** Impacts of air pollution on NPP in the whole of China. Results shown are combined effects of aerosol and $O_3$ on the summer NPP in (a) 2010, (b) 2030 with CLE scenario, and (c) 2030 with MTFR scenario. Results for the top panels do not include aerosol indirect effects (AIE) but do include the meteorological response to aerosol direct radiative effects. The average NPP perturbation over the box domain in (a) is shown in the bracket of each title. The perturbations to annual total NPP by aerosol, $O_3$, and their sum over the whole China are shown in (d-f) for different periods, with (right) and without (left) inclusion of AIE. Damages by $O_3$ are averaged for low and high sensitivities with error bars indicating ranges. The percentage changes are calculated based on NPP without AIE. Simulations are performed with the ModelE2-YIBs model. Only the significant changes ($p < 0.05$) are shown in (a-c).

1086
1087