# Peer review of "Ozone and haze pollution weakens net primary productivity in China"

_Atmospheric Chemistry and Physics, 2016_

## Referee Comment (RC1) · W.J. Collins (Referee) · 11 Jan 2017

The paper by Yue et al. is a valuable assessment of effects of ozone and aerosol pollution on NPP over China. In particular this is the first time the aerosol contribution has been examined in such detail. The modelled ozone damage is compared against field measurements. This paper should certainly be published in ACP, however some revision is needed as described below.

The authors should make it clearer how much of the impact of ozone and aerosols on NPP is natural and how much anthropogenic. The headline numbers are referred to as due to air pollution, but presumably there would be effects on NPP due to natural ozone. Two extra runs G10NATLO3 and G10NATHO3 would provide the required data for this. The authors assert that since the NPP effect is small below 40 ppb the no ozone and

natural ozone simulations are equivalent, but the authors need to demonstrate this with these two extra runs.

When analysing the meteorological changes the authors only show the impacts over China. In their global model set up the aerosols will change globally and affect global circulation (even with fixed SST). The authors should therefore show global maps corresponding to figures 7 and S8 in the supplement. One feature of perturbing aerosols in fixed-SST simulations is that there are large changes in the land-sea temperature contrast and hence artificial changes in circulation patterns. The resulting meteorological changes over China will therefore be a combination of locally driven effects (such as change in radiation and hence evaporation) and regional-globally driven effects (such as changes in rainfall and hence soil water). This seems to be particularly apparent in the AIE simulations where the patterns of changes in precipitation and soil water bear no relation to the changes in aerosol. The soil moisture changes dominate the aerosol impacts on NPP and I am not convinced these can be attributed to the aerosol changes. The changes in PAR and surface temperature can be much more readily linked physically to the changes in aerosol, therefore the authors should exclude the soil moisture changes from their analysis in table 2.

More explanation of table 2 is needed. What simulations are compared against what to derive the answers? How are the uncertainties derived? – presumably they are interannual variability, but different sets of annually-varying data are used for the online and offline calculations.

Specific points

Page 2, lines 47-53: Uncertainties (including the range between high and low sensitivity) need to be included here.

Page 2, line 55: suggest to replace "will not alleviate" with "will be further increased".

Page 4, line 93: "not been properly validated" – The authors need to be more explicit

about exactly what deficiencies the previous studies had.

Page 7, line 186: The authors should clarify that they are referring to 2010 emissions.

Page 7, line 201: How much does biomass burning contribute to the emissions? Are they considered natural (in G10NATxxx)?

Page 7, lines 205-210: The authors need to describe how the changes in natural emissions are determined.

Page 8, line 217. I suggest including the table of simulations in the main text rather than the supplement.

Page 8, line 232. The authors should list the changes in WMGHG from 2010 to 2030 (at least CO2 and methane).

Page, 9 section 2.4.3. This section isn't clear about how the meteorological changes are applied to the offline model. Are they applied as an average (of the last 15 years) of the table S2 simulations; or are individual years from the table S2 simulations used as input. If the former: why is there any variability in the offline output? If the latter: the last 3 years could be strongly influenced by interannual variability.

Page 10, lines 298-299. The agreement in figure 3 doesn't suggest that the "Evaluations at rural sites better match the observations". The correlation is no better than for all sites, and by eye only the summer points look to show any correlation at all.

Page 12, line 346-348. The online model presumably allows the g_s changes to feed back on the ozone concentrations, which should increase them. Therefore it might be expected that the online model would show more ozone damage. The authors should compare the g_s and surface ozone concentration changes between the online and offline models.

Page 12, lines 348-349. Are the authors saying they have carried out a G10NATLO3 simulation, and the NPP change (compared to G10NATNO3) is identically zero everywhere? If so, this needs to be explained more clearly. If not, then the authors need to be clearer about the evidence they have that the zero anthropogenic emissions show no damage. The damage functions in fig 5 don't go exactly to zero at 40 ppb.

Page 12, line 352. The uncertainty here also needs to include the uncertainty in plant sensitivity (i.e. the range from high to low). Technically you should refer to the "central value" between high and low, rather than "average".

Page 12, line 362. Have the authors checked whether the absolute relative humidity is affected, i.e. whether the relative humidity change is purely due to the decreased temperature.

Page 12, lines 363-364. There are a lot of statements presented here without any evidence. The authors have not shown strengthened plant transpiration and have not demonstrated that any increase in RH is due to this (rather than simply decreased temperatures or increased horizontal moisture transport). Similarly the authors have not shown diagnostics demonstrating a direct causal chain between transpiration and precipitation or cloud cover. Both of these could instead be due to changes in circulation patterns.

Page 12, lines 367-369. Again no evidence is presented that the decrease in summer precipitation is due to a reduction in the cloud droplet size. Ultimately precipitation is driven by moisture convergence.

Page 13, lines 382-384. This sentence wasn't very clear. Is it referring to changes in heterotrophic respiration? If so, it should be said explicitly.

Page 13, line 395. There doesn't seem to be any change in soil water in the North China Plain (fig S8f) in the same region where NPP decreases in fig S9d.

Page 14, line 404. Is the agreement between the offline and online $O_3$ inhibition true as a geographical pattern as well as the China total?

Page 14, line 406-407. Explain that the range quoted is for no AIE compared to AIE.

[Figure]

Uncertainties should also be quoted and include the range between high and low sensitivity.

Page 14, line 427. What is the change in methane in 2030?

Page 15, lines 434-436. It would be useful to be told the change in g_s between 2010 and 2030.

Page 15, line 436. Need to include the high-low sensitivity range here.

Page 15, line 441. Need to include the high-low sensitivity range here.

Figure 3. The right hand plot needs a legend to explain the colours.

Figures 4, 6, 8, 9, 10, S5, S7, S9 : The south east China box should be shown in every panel.

Figure 4. The key for blue and black dots should be provided within the graphs.

Figure 5. The key for colours should be provided in the graphs. It would be useful to provide the letter keys within table S1. A different key may be better as there a several authors starting with "Z".

Figure 8. Clarify here and/or in the text that the PAR changes include the DRF.

Figure S3. The key for colours should be provided in the graphs.

Figure S5. The colour scale for percentages should use the red colours for all the positive values, and blue only if there are negative ones, otherwise use the same colours as for the absolute values.

---

## Referee Comment (RC2) · Anonymous Referee #2 · 11 Jan 2017

This study explores the impact of air pollution on crop production, with a specific focus on China. This is a nice study and in many ways ambitious in scope, though it builds on a series of YIBs model developments described in previous literature. This is a great application of coupling atmospheric chemistry and biosphere modeling and in general I found the paper was well executed. I suggest a little more work to clarify the details behind these results, but after these minor corrections, the paper should be in good shape for publication in ACP.

Specific Comments

1. The paper would benefit from a clearer distinction/discussion of impacts attributed to meteorology feedbacks from PM & O3 forcing vs. aerosol indirect effects. The former are referred to as "direct effects" though they are in fact meteorological feedbacks. In

general, it would be helpful if the authors provided a clearer quantification of these specific effects and the model simulations used to assess them.

2. The meteorological & hydrological responses presented primarily in 3.3 should include some standard deviation numbers since multiple years of simulation were run to assess natural variability. Are the changes in soil moisture and precipitation significant?

3. The paper needs a more consistent time-scale. The overall results are presented as annual, however all the figures (except Fig 10) show summertime results only. The authors should either include evaluation for all seasons (or annual means), or present the final results only for summer. As is, the reader cannot judge model skill or response for other seasons.

4. The paper should discuss the potential implications of the high bias in simulated diffuse fraction and potentially in O3 (the evaluation of simulated O3 is mixed).

Details

1. Line 71: typo "meteorology, and clouds."

2. Line 90: need to define the square brackets in [O3]

3. Line 93-94: language "less well understood"

4. Line 149-150: not quite true, the CLM includes more PFTs, this should be clarified here.

5. Line 188: this is a large difference in NH3 emissions, do the authors know why the inventories differ?

6. Line 202-203: do these changes in biomass burning emissions seem realistic?

7. Lines 205-207: are these natural emissions simulated online or specified? Are there appropriate references that could be cited for this?

8. Lines 208-209: Please explain why isoprene emissions increase and monoterpene

emissions decrease (text later indicates that land cover is fixed)

9. Section 3.1.1 & Figure 1: Please discuss the spatial differences between observed and simulated GPP/NPP.

10. Line 282: Is R=0.86 a typo? Figure 1 suggests this should be 0.75

11. Figure 2 caption: should include years

12. Section 3.1.2 & Figure 2: Please briefly discuss where the model is too high and too low and what species might contribute to these biases. Also quantify the last sentence (line 298-299)

13. Line 308: "diffuse fraction agree" – this is incorrect. The simulation appears biased quite high in some regions. Please correct.

14. Section 4.2 should also acknowledge that the response of the hydrological cycle to aerosols is also a major source of uncertainty.

---

## Author Comment (AC1) · 27 Feb 2017

**Referee 1: Prof. William Collins**

We are grateful to Prof. William Collins for his time and energy in providing helpful comments and guidance that have improved the manuscript. In this document, we describe how we have addressed the reviewer's comments. Referee comments are shown in black italics and author responses are shown in blue regular text.

The paper by Yue et al. is a valuable assessment of effects of ozone and aerosol pollution on NPP over China. In particular this is the first time the aerosol contribution has been examined in such detail. The modelled ozone damage is compared against field measurements. This paper should certainly be published in ACP, however some revision is needed as described below.

The authors should make it clearer how much of the impact of ozone and aerosols on NPP is natural and how much anthropogenic. The headline numbers are referred to as due to air pollution, but presumably there would be effects on NPP due to natural ozone. Two extra runs G10NATLO3 and G10NATHO3 would provide the required data for this. The authors assert that since the NPP effect is small below 40 ppb the no ozone and natural ozone simulations are equivalent, but the authors need to demonstrate this with these two extra runs.

→ We performed two extra runs, G10NATLO3\_OFF and G10NATHO3\_OFF, which consider offline vegetation damages due to  $O_3$  from natural emissions alone. We added a new Figure S7 to compare the GPP reductions caused by  $O_3$  with and without anthropogenic emissions. It shows that  $O_3$  from natural sources has trivial impacts on GPP.

In the Methods section, we revised the text to introduce the two additional experiments: "To compare the differences between online and offline O3 damage, we perform four additional simulations which do not account for the feedbacks of O3-induced changes in biometeorology, plant growth, and ecosystem physiology. Two simulations, G10ALLHO3\_OFF and G10ALLLO3\_OFF, include both natural and anthropogenic emissions. The other two, G10NATHO3\_OFF and G10NATHO3\_OFF, include natural emissions alone." (Lines 253-259)

In the Results section, we describe the findings from the extra runs: "Sensitivity simulations with zero anthropogenic emissions show almost no  $O_3$  damage (Fig. S7), because the  $[O_3]$  exposure from natural sources alone is usually lower than the threshold level of 40 ppbv below which the damage for most PFTs is limited (Fig. 5)." (Lines 402-405)

When analysing the meteorological changes the authors only show the impacts over China. In their global model set up the aerosols will change globally and affect global circulation (even with fixed SST). The authors should therefore show global maps corresponding to figures 7 and S8 in the supplement. One feature of perturbing aerosols in fixed-SST simulations is that there are large changes in the land-sea temperature contrast and hence artificial changes in circulation patterns. The resulting meteorological changes over China will therefore be a combination of locally driven effects (such as change in radiation and hence evaporation) and regional-globally driven effects (such as changes in rainfall and hence soil water). This seems to be particularly apparent in the AIE simulations where the patterns of changes in precipitation and soil water bear no relation to the changes in aerosol. The soil moisture changes dominate the aerosol impacts on NPP and I am not convinced these can be attributed to the aerosol changes. The changes in PAR and surface temperature can be much more readily linked physically to the changes in aerosol, therefore the authors should exclude the soil moisture changes from their analysis in table 2.

We do agree that the aerosol-induced changes in meteorology over China are a combination of local and remote effects, but we assert that the changes (local and remote) are all ultimately attributed to the aerosol radiative perturbation. For example, the regional soil moisture changes are absolutely caused by the aerosol radiative perturbations, but may be more linked to a regionally-globally driven dynamical mechanism rather than the reduction in local downward shortwave. The main goal of this study is to investigate the terrestrial biospheric response to air pollution in China. Diagnosing long range dynamical mechanisms is out of scope of this study, especially because the aerosol impacts are so much smaller than the ozone impacts, this specific study will not gain from an explicit description of the multi-scale dynamical mechanisms that drive the regional meteorological changes. Furthermore, the paper is already quite long with 10 figures and supporting information, and global maps obscure and make it difficult for readers to see the regional signals in China. Therefore, we decide not to add further figures of global maps corresponding to figures 7 and S9 in the supplement. In Section 4.1, we already include a comparison of the simulated aerosol impacts on meteorology and surface climate against existing published estimates and a discussion of their realism and uncertainties. Similarly, we select to retain the soil moisture results in Table 3 (original Table 2). Soil moisture dominates the NPP response in only 2 out of 6 scenarios/cases. Even if the soil wetness changes occur in part through more long range dynamical mechanisms triggered by the aerosol radiative perturbations, it is important to highlight the biospheric sensitivity to changes in this driver, oftentimes ignored/neglected in aerosol-carbon-climate studies. For example, precipitation controls GPP in more than 40% of vegetated land (Beer et al., Science, 2010).

Following Prof. Collin's suggestions, we make several modifications.

We add: "The resulting meteorological changes over China are a combination of locally driven effects (such as changes in radiation and hence temperature) and regional-globally driven effects (such as changes in rainfall and hence soil water)." (Lines 431-434).

We clarify: "Aerosol-induced impacts on precipitation and soil moisture are not statistically significant over the regionally averaged domain (Tables S5 and S6). However, for the 2010 and 2030 CLE cases with AIE, 2 out of 6 scenarios, the aerosol-induced impact on soil moisture dominates the total NPP response (Table 3)." (Lines 626-629)

We emphasize: "our estimate of NPP response to aerosol effects, with or without AIE, is secondary in magnitude compared to the O3 vegetation damage." (Lines 633-634)

More explanation of table 2 is needed. What simulations are compared against what to derive the answers? How are the uncertainties derived? – presumably they are interannual variability, but different sets of annually-varying data are used for the online and offline calculations.

→ We added more descriptions in the footnote of Table 3 (original Table 2) to explain how we derive those numbers from different simulations. We clarified in the section 2.4.1 about the uncertainties: "The full list of simulations in Table 2 offers assessment of uncertainties due to interannual climate variability, emission inventories (CLE or MTFR), O3 damage sensitivities (low to high), and aerosol climatic effects (direct and indirect). Uncertainties calculated based on the interannual climate variability in the model are indicated using the format 'mean ± one standard deviation'. Other sources of uncertainty are explicitly stated." (Lines 276-281).

The interannual climate variability from online and offline simulations are calculated using different time periods. We clarified in the section 2.4.3 as follows: "Uncertainties due to interannual climate variability in the model are calculated using different time periods for the online (15 years, Table 2) and offline (10 years, Table S3) runs." (Lines 311-313).

**Specific points**

*Page 2, lines 47-53: Uncertainties (including the range between high and low sensitivity) need to be included here.*

→ We included uncertainties due to  $O_3$  sensitivity: "In the present day,  $O_3$  reduces annual NPP by 0.6 Pg C (14%) with a range from 0.4 Pg C (low  $O_3$  sensitivity) to 0.8 Pg C (high  $O_3$  sensitivity). In contrast, aerosol direct effects increase NPP by 0.2 Pg C (5%) through the combination of diffuse radiation fertilization, reduced canopy temperatures, and reduced evaporation leading to higher soil moisture. Consequently, the net effects of  $O_3$  and aerosols decrease NPP by 0.4 Pg C (9%) with a range from 0.2 Pg C (low  $O_3$  sensitivity) to 0.6 Pg C (high  $O_3$  sensitivity). However, precipitation inhibition from combined aerosol direct and indirect effects reduces annual NPP by 0.2 Pg C (4%), leading to a net air pollution suppression of 0.8 Pg C (16%) with a range from 0.6 Pg C (low  $O_3$  sensitivity) to 1.0 Pg C (high  $O_3$  sensitivity)." (Lines 47-56)

Page 2, line 55: suggest to replace "will not alleviate" with "will be further increased".

 $\rightarrow$  Corrected as suggested.

Page 4, line 93: "not been properly validated" – The authors need to be more explicit about exactly what deficiencies the previous studies had.

→ We clarified the sentence as follows: "Previous regional modeling ... does not always take advantage of valuable observations to calibrate GPP-O3 sensitivity coefficients for the China domain and typically the derived results have not been properly validated." (Lines 94-97)

Page 7, line 186: The authors should clarify that they are referring to 2010 emissions.

→ We clarified as follows: "Inter-comparison of present-day (the year 2010) emissions (Fig. S2) shows ..." (Lines 190-191)

Page 7, line 201: How much does biomass burning contribute to the emissions? Are they considered natural (in G10NATxxx)?

→ We clarified as follows: "Biomass burning emissions, adopted from the IPCC RCP8.5 scenario (van Vuuren et al., 2011), are considered as anthropogenic sources because most fire activities in China are due to human-managed prescribed burning. Compared with the GAINs inventory, present-day biomass burning is equivalent to  $\leq 1\%$  of the emissions for NOx, SO2, and NH3, 1.6% for BC, 3.0% for CO, and 9.6% for OC." (Lines 213-217)

Page 7, lines 205-210: The authors need to describe how the changes in natural emissions are determined.

 $\rightarrow$  We add a detailed description of the climate-sensitive natural emission sources:

"The model represents climate-sensitive natural precursor emissions of lightning  $NO_x$ , soil NOx and biogenic volatile organic compounds (BVOCs) (Unger and Yue, 2014). Future 2030 changes in these natural emissions are small compared to the anthropogenic emission changes. Interactive lightning NOx emissions are calculated based on the climate model's moist convection scheme that is used to derive the total lightning and the cloud-to-ground lightning frequencies (Price et al., 1997; Pickering et al., 1998; Shindell et al., 2013). Annual average lightning NOx emissions over China increase by 4% between 2010 and 2030. Interactive biogenic soil NOx emission is parameterized as a function of PFT-type, soil temperature, precipitation (including pulsing events), fertilizer loss, LAI, NOx dry deposition rate, and canopy wind speed (Yienger and Levy, 1995). Annual average biogenic soil NOx emissions increase by only 1% over China between 2010 and 2030. Leaf isoprene emissions are simulated using a biochemical model that depends on the electron transport-limited photosynthetic rate, intercellular CO2, canopy temperature, and atmospheric CO2 (Unger et al., 2013). Leaf monoterpene emissions depend on canopy temperature and atmospheric CO2 (Unger and Yue, 2014). Annual average isoprene emission in China increases by 5% (0.39 TgC/yr) between 2010 and 2030 in response to enhanced GPP and temperature that offset the effects of CO2inhibition. Monoterpene emissions decrease by 5% (-0.25 Tg C) between 2010 and 2030 because CO2-inhibition outweighs the effects of increased warming." (Lines 220-238).

Page 8, line 217. I suggest including the table of simulations in the main text rather than the supplement.

 $\rightarrow$  We have moved Table S2 into the main text as suggested (now Table 2).

Page 8, line 232. The authors should list the changes in WMGHG from 2010 to 2030 (at least CO2 and methane).

→ We clarified as follows: "Well-mixed GHG concentrations are also adopted from the RCP8.5 scenario, with CO2 changes from 390 ppm in 2010 to 449 ppm in 2030, and CH4 changes from 1.779 ppm to 2.132 ppm." (Lines 264-266)

Page, 9 section 2.4.3. This section isn't clear about how the meteorological changes are applied to the offline model. Are they applied as an average (of the last 15 years) of the table S2 simulations; or are individual years from the table S2 simulations used as input. If the former: why is there any variability in the offline output? If the latter: the last 3 years could be strongly influenced by interannual variability.

→ We add one sentence to explain how the meteorological changes are applied to the offline model: "For these simulations, the month-to-month meteorological perturbations caused by aerosols are applied as scaling factors on the baseline forcing." (Lines 307-309) We actually run the offline simulations for 15 years, with the last 10 years used for the analyses. The paper has been updated for several versions. In the first version, we ran the offline simulations for only 10 years and did not show the uncertainties in the Table because the time period was short. In the latest version, we have already re-ran all simulations to 15 years and calculated uncertainties for Table 3. However, we had omitted to update the corresponding manuscript text.

Page 10, lines 298-299. The agreement in figure 3 doesn't suggest that the "Evaluations at rural sites better match the observations". The correlation is no better than for all sites, and by eye only the summer points look to show any correlation at all.

→ We revise the text as follows: "Evaluations at rural sites (Table S4), which represent the major domain of China, show a mean bias of -5% (Fig. 3). The magnitude of such bias is much lower than the value of 42.5% for the comparisons at urban-dominant sites (Fig. 2f)." (Lines 346-348)

Page 12, line 346-348. The online model presumably allows the  $g_s$  changes to feed back on the ozone concentrations, which should increase them. Therefore it might be expected that the online model would show more ozone damage. The authors should compare the g s and surface ozone concentration changes between the online and offline models.

→ The online model does allow  $g_s$  changes to feed back onto the atmospheric composition. However, our model does not show significant O3 concentration feedbacks at the current level of vegetation and  $g_s$  damage, likely because of multiple offsetting

influences on chemistry. We have now included  $g_s$  changes in a revised Figure S6 and described them in the text as follows: "At the same time, the O3-induced reductions in stomatal conductance (Fig. S6a) can increase canopy temperature and inhibit plant transpiration, leading to surface warming (Fig. S6b), dry air (Fig. S6c), and rainfall deficit (Fig. S6d)." (Lines 395-398)

Page 12, lines 348-349. Are the authors saying they have carried out a G10NATLO3 simulation, and the NPP change (compared to G10NATNO3) is identically zero everywhere? If so, this needs to be explained more clearly. If not, then the authors need to be clearer about the evidence they have that the zero anthropogenic emissions show no damage. The damage functions in fig 5 don't go exactly to zero at 40 ppb.

→ We performed two additional sensitivity runs with the climate model ModelE2-YIBs, G10NATLO3\_OFF and G10NATHO3\_OFF, to examine damages by natural O3. We found negligible instead of zero impacts of background O3 from natural precursor sources in China. We show the results in the new Figure S7. In the main text, we clarified as follows: "Sensitivity simulations with zero anthropogenic emissions show almost no O3 damage (Fig. S7), because the [O3] exposure from natural sources alone is usually lower than the threshold level of 40 ppbv below which the damage for most PFTs is limited (Fig. 5)." (Lines 402-405)

Page 12, line 352. The uncertainty here also needs to include the uncertainty in plant sensitivity (i.e. the range from high to low). Technically you should refer to the "central value" between high and low, rather than "average".

→ We clarified as follows: "Our results indicate that present-day surface  $O_3$  causes strong inhibitions on total NPP in China, ranging from  $0.43 \pm 0.12$  Pg C yr-1 with low sensitivity to  $0.76 \pm 0.15$  Pg C yr-1 with high sensitivity (Table 3). The central value of NPP reduction by  $O_3$  is  $0.59 \pm 0.11$  Pg C yr-1, assuming no direct impacts of  $O_3$  on plant respiration." (Lines 405-409)

Page 12, line 362. Have the authors checked whether the absolute relative humidity is affected, i.e. whether the relative humidity change is purely due to the decreased temperature.

→ We plotted changes in stomatal conductance and specific humidity (the figure below). It shows that specific humidity changes little. As a result, most of the changes in RH are driven by lower saturation vapor pressure. In the text, we clarified as follows: "Reduced insolation decreases summer surface temperature by 0.63°C in the East, inhibiting evaporation but increasing relative humidity (RH) due to the lower saturation vapor pressure (Table S5). These feedbacks combine to stimulate photosynthesis (Fig. 8a), which, in turn, strengthens plant transpiration (not shown)." (Lines 417-421).

(a) Relative changes in stomatal conductance (%) (b) Relative changes in specific humidity (%)

Page 12, lines 363-364. There are a lot of statements presented here without any evidence. The authors have not shown strengthened plant transpiration and have not demonstrated that any increase in RH is due to this (rather than simply decreased temperatures or increased horizontal moisture transport). Similarly the authors have not shown diagnostics demonstrating a direct causal chain between transpiration and precipitation or cloud cover. Both of these could instead be due to changes in circulation patterns.

→ We agree that the original statements were rather terse. As shown above, the increase in transpiration (left panel of the above figure) does not increase specific humidity (right panel). Advection and convergence may alter the local moisture budget. In the revised text, we clarify as follows: "Atmospheric circulation and moisture convergence are also altered due to the pollution-vegetation-climate interactions, resulting in enhanced precipitation (Fig. 7b) and cloud cover (Fig. 7d)." (Lines 421-423)

Page 12, lines 367-369. Again no evidence is presented that the decrease in summer precipitation is due to a reduction in the cloud droplet size. Ultimately precipitation is driven by moisture convergence.

→ The precipitation changes are due to a combination of altered cloud microphysics and atmospheric circulation patterns for which we have no way of disentangling. We modify the statement as follows: "Inclusion of AIE results in distinct climatic feedbacks (Fig. S9). Summer precipitation decreases by 0.9 mm day-1 (13%), leading to a 3% decline in soil moisture (Table S6)." (Lines 424-426)

Page 13, lines 382-384. This sentence wasn't very clear. Is it referring to changes in heterotrophic respiration? If so, it should be said explicitly.

 $\rightarrow$  We have removed the sentence about the impact of phenological changes. Quantification of such effects requires additional simulations, which are out of the scope of this study. Page 13, line 395. There doesn't seem to be any change in soil water in the North China Plain (fig S8f) in the same region where NPP decreases in fig S9d.

(f) Relative changes in soil water (%)

→ The results shown in Fig. S9f (original Fig. S8f) are based on last 15-year simulations (years 6-20). However, results of offline simulations are based on 10-year meteorology (years 6-15). We show the changes in soil moisture for years 6-15 in the above. We can see that the pattern of soil water deficit in the above figure matches NPP changes in Fig. S10d (original Fig. S9d). The changes of soil water in North China Plain are not statistically significant in both the 15-year and 10-year simulations. The lack of significance may cause the inconsistency of NPP changes, i.e., the pattern of Fig. S10d, between the 10-year and 15-year simulations. However, the main conclusion that soil moisture plays the domain role in the NPP responses remains correct because NPP changes by temperature and radiation are far smaller than that by soil moisture. In the text, we added the following statement to remind readers about the discrepancies between online and offline simulations: "Uncertainties due to interannual climate variability in the model are calculated using different time periods for the online (15 years, Table 2) and offline (10 years, Table S3) runs." (Lines 311-313)

**Page 14, line 404. Is the agreement between the offline and online O3 inhibition true as a geographical pattern as well as the China total?**

→ Yes. We show both the online and offline damage below. Although we can see some differences at the individual grid cell level, the spatial patterns are quite similar between the online and offline runs. We do not present the figure in the paper because it is already busy with 10 Figures and supporting information. Instead, we state: "Application of ModelE2-YIBs that allows for these feedbacks gives an O3-induced damage to annual GPP of 10.7%, a similar level to the damage computed in YIBs offline. The spatial pattern of the online O3 inhibition also resembles that of offline damages (not shown)." (Lines 399-402)

---

## Author Comment (AC2) · 27 Feb 2017

**Referee 2**

We are grateful to the reviewer for their time and energy in providing helpful comments and guidance that have improved the manuscript. In this document, we describe how we have addressed the reviewer's comments. Referee comments are shown in black italics and author responses are shown in blue regular text.

*This study explores the impact of air pollution on crop production, with a specific focus on China. This is a nice study and in many ways ambitious in scope, though it builds on a series of YIBs model developments described in previous literature. This is a great application of coupling atmospheric chemistry and biosphere modeling and in general I found the paper was well executed. I suggest a little more work to clarify the details behind these results, but after these minor corrections, the paper should be in good shape for publication in ACP.*

**Specific Comments**

*1. The paper would benefit from a clearer distinction/discussion of impacts attributed to meteorology feedbacks from PM & O3 forcing vs. aerosol indirect effects. The former are referred to as "direct effects" though they are in fact meteorological feedbacks. In general, it would be helpful if the authors provided a clearer quantification of these specific effects and the model simulations used to assess them.*

→ We appreciate that the terminology may be a little challenging in multidisciplinary studies, but we believe that our decisions have made the impacts as clear as possible. We have adopted the use of "direct" and "indirect" as exactly used in the IPCC assessments because these terms are widely used in the aerosol-climate community. The direct and indirect aerosol effects are both associated with meteorological feedbacks. Throughout the manuscript, we emphasize when we are referring to feedbacks and whether they derive from aerosol direct and/or indirect effects.

*2. The meteorological & hydrological responses presented primarily in 3.3 should include some standard deviation numbers since multiple years of simulation were run to assess natural variability. Are the changes in soil moisture and precipitation significant?*

→ We have separated the original Table S6 into two Tables, with S5 for annual statistics and S6 for summer statistics. Each Table includes the mean changes and one standard deviation (brackets) indicating the uncertainties.

In Section 3.3, we have added following statement to emphasize that the changes in hydrological fields have large uncertainties: "Compared to aerosol-induced perturbations in radiation and temperature, responses in hydrological variables (e.g. precipitation and soil moisture) are usually statistically insignificant on the domain average due to the large relative interannual climate variability (Tables S5 and S6)." (Lines 428-431).

*3. The paper needs a more consistent time-scale. The overall results are presented as annual, however all the figures (except Fig 10) show summertime results only. The authors should either include evaluation for all seasons (or annual means), or present the final results only for summer. As is, the reader cannot judge model skill or response for other seasons.*

→ The reason why our analyses and the Figures focus on the summer is that both GPP/NPP and air pollution (especially $O_3$) reach maximum at this season. The largest interactions between carbon flux and air pollution are found for this season. It is not a contradiction to show Figures on the summer average and provide annual average impacts because the carbon loss in summer largely dominates the annual total. We found that, for $O_3$ damages, "about 61% of such inhibition occurs in summer, when both photosynthesis and [$O_3$] reach maximum of the year." (Lines 409-410). For the combined $O_3$ and aerosol effects, "a dominant fraction (60% without AIE and 52% with AIE) of the reduced carbon uptake occurs in summer, when both NPP and [$O_3$] reach maximum of the year." (Lines 474-476). We also elect to present and summarize the annual average results to the reader for consistency with regional carbon budget studies. Having the annual average values easily available facilitates comparison with other carbon flux impacts and carbon emissions. For example, we found that: "the combined effects of $O_3$ and aerosols (Table 3) decrease total NPP in China by 0.39 (without AIE) to 0.80 Pg C $yr^{-1}$ (with AIE), equivalent to 9-16% of the pollution-free NPP and 16-32% of the total anthropogenic carbon emissions". (Lines 469-471)

*4. The paper should discuss the potential implications of the high bias in simulated diffuse fraction and potentially in O3 (the evaluation of simulated O3 is mixed).*

→ We added following statements to discuss the implications of biases in diffuse fraction and O3: "Predicted [$O_3$] is largely overestimated at urban sites but exhibits reasonable magnitude at rural sites (Figs 2 and 3). Measurements of background [$O_3$] in China are limited both in space and time, restricting comprehensive validation of [$O_3$] and the consequent estimate of $O_3$ damages on the country level." (Lines 595-598)

"The model overestimates diffuse fraction in China (Fig. 4), likely because of simulated biases in clouds. Previously, we improved the prediction of diffuse fraction in China using observed cloud profiles for the region (Yue and Unger, 2017). Biases in simulated AOD and diffuse fraction introduce uncertainties in the aerosol DRF especially in the affected localized model grid cells. Yet, averaged over the China domain, our estimate of NPP change by aerosol DRF (0.09 Pg C $yr^{-1}$) is consistent with the previous assessment in Yue and Unger (2017) (0.07 Pg C $yr^{-1}$)." (Lines 619-625)

*Details*

*1. Line 71: typo "meteorology, and clouds."*

→ Corrected as suggested.

*2. Line 90: need to define the square brackets in [O3]*

→ We added the following definition: "…O₃ concentrations ([$O_3$])" (Line 94).

*3. Line 93-94: language "less well understood"*

→ Corrected as suggested.

*4. Line 149-150: not quite true, the CLM includes more PFTs, this should be clarified here.*

→ The 8 PFTs used in climate model ModelE2-YIBs are aggregated from a land cover data set with 16 PFTs, which are used by the CLM model. We clarified as follows: "For both global and regional simulations, 8 plant functional types (PFTs) are considered (Fig. S1). This land cover is aggregated from a dataset with 16 PFTs, which are derived using retrievals …. The same vegetation cover with 16 PFTs is used by the Community Land Model (CLM)" (Lines 150-155).

5. Line 188: this is a large difference in NH3 emissions, do the authors know why the inventories differ?

→ We clarify as follows: "The discrepancies among different inventories emerge from varied assumptions on the stringency and effectiveness of emission control measures. While the GAINS 2010 ammonia emissions from China are larger than the RCP8.5 and HTAP emissions as shown in Fig. S2, they are close in magnitude to the year 2010 emissions of 13.84 Tg yr⁻¹ estimated by the Regional Emission inventory in ASia (REAS, http://www.nies.go.jp/REAS/)." (Lines 193-198)

*6. Line 202-203: do these changes in biomass burning emissions seem realistic?*

→ The reviewer raises an interesting and provocative question. The future changes in biomass burning in China are small, and that is indeed realistic based on current understanding of fire activity in China today. For example, wildfire activity is limited in China today. We state in the manuscript: "Biomass burning emissions, adopted from the IPCC RCP8.5 scenario (van Vuuren et al., 2011), are considered as anthropogenic sources because most fire activities in China are due to human-managed prescribed burning. Compared with the GAINs inventory, present-day biomass burning is equivalent to <1% of the emissions for NOₓ, SO₂, and NH₃, 1.6% for BC, 3.0% for CO, and 9.6% for OC." (Lines 213-217)

*7. Lines 205-207: are these natural emissions simulated online or specified? Are there appropriate references that could be cited for this?*

→ We add a detailed description of the climate-sensitive natural emission sources:

"The model represents climate-sensitive natural precursor emissions of lightning $NO_x$, soil $NO_x$ and biogenic volatile organic compounds (BVOCs) (Unger and Yue, 2014). Future 2030 changes in these natural emissions are small compared to the anthropogenic emission changes. Interactive lightning $NO_x$ emissions are calculated based on the climate model's moist convection scheme that is used to derive the total lightning and the cloud-to-ground lightning frequencies (Price et al., 1997; Pickering et al., 1998; Shindell et al., 2013). Annual average lightning $NO_x$ emissions over China increase by 4% between 2010 and 2030. Interactive biogenic soil $NO_x$ emission is parameterized as a function of PFT-type, soil temperature, precipitation (including pulsing events), fertilizer loss, LAI, $NO_x$ dry deposition rate, and canopy wind speed (Yienger and Levy, 1995). Annual average biogenic soil $NO_x$ emissions increase by only 1% over China between 2010 and 2030. Leaf isoprene emissions are simulated using a biochemical model that depends on the electron transport-limited photosynthetic rate, intercellular $CO_2$, canopy temperature, and atmospheric $CO_2$ (Unger et al., 2013). Leaf monoterpene emissions depend on canopy temperature and atmospheric $CO_2$ (Unger and Yue, 2014). Annual average isoprene emission in China increases by 5% (0.39 TgC/yr) between 2010 and 2030 in response to enhanced GPP and temperature that offset the effects of $CO_2$-inhibition. Monoterpene emissions decrease by 5% (-0.25 Tg C) between 2010 and 2030 because $CO_2$-inhibition outweighs the effects of increased warming." (Lines 220-238).

*8. Lines 208-209: Please explain why isoprene emissions increase and monoterpene emissions decrease (text later indicates that land cover is fixed)*

→ Please see above response to Point (7).

*9. Section 3.1.1 & Figure 1: Please discuss the spatial differences between observed and simulated GPP/NPP.*

→We add the following information to Section 3.1.1: "For GPP, prediction in the summer overestimates by 6.2% over the southern coast (< 28°N), but underestimates by 23.7% over the North China Plain (NCP, [32-40°N, 110-120°E]). Compared with the MODIS data product, predicted summer NPP is overall overestimated by 20.6% in China (Fig. 1f), with regional biases of 40.0% in the southern coast, 51.2% in the NCP, and 38.7% in the Northeast (> 124°E)." (Lines 327-331)

*10. Line 282: Is R=0.86 a typo? Figure 1 suggests this should be 0.75*

→ The R=0.86 is for the annual GPP as shown in Figure S4. We have indicated both Figure 1 and Figure S4 in the text. (Line 325)

*11. Figure 2 caption: should include years*

→ We added the information of years as follows: "…observations from (b) the satellite retrieval of the MODIS (averaged for 2008-2012), and (e) and (h) measurements from 188 ground-based sites (at the year 2014)" (Lines 959-961)

*12. Section 3.1.2 & Figure 2: Please briefly discuss where the model is too high and too low and what species might contribute to these biases. Also quantify the last sentence (line 298-299)*

→ We describe the AOD biases as follows: "Predicted AOD also reproduces the observed spatial pattern, but underestimates the high center in NCP by 24.6%." (Lines 339-340)

In the Discussion Section 4.2, we explain the cause of AOD biases: "Simulated surface $PM_{2.5}$ is reasonable but AOD is underestimated in the North China Plain (Fig. 2), likely because of the biases in aerosol optical parameters. Using a different set of optical parameters, we predicted much higher AOD that is closer to observations with the same aerosol vertical profile and particle compositions (Yue and Unger, 2017)." (Lines 615-619)

We revise the text as follows: "Evaluations at rural sites (Table S4), which represent the major domain of China, show a mean bias of -5% (Fig. 3). The magnitude of such bias is much lower than the value of 42.5% for the comparisons at urban-dominant sites (Fig. 2f)." (Lines 346-348)

*13. Line 308: "diffuse fraction agree" – this is incorrect. The simulation appears biased quite high in some regions. Please correct.*

→ We corrected the sentence: "Simulated diffuse fraction reproduces observed spatial pattern with high correlation coefficient ($r = 0.74$, $p < 0.01$), though it is on average 25.2% larger than observations (Figs 4d-4f). Such bias is mainly attributed to the overestimation in the North and Northeast. For the southeastern region, where high values of GPP dominate (Fig. 1), predicted diffuse fraction is in general within the 10% difference from the observations." (Lines 357-362)

*14. Section 4.2 should also acknowledge that the response of the hydrological cycle to aerosols is also a major source of uncertainty.*

→ We revised the text to acknowledge the uncertainty as follows: "Aerosol-induced impacts on precipitation and soil moisture are not statistically significant over the regionally averaged domain (Tables S5 and S6). However, for the 2010 and 2030 CLE cases with AIE, 2 out of 6 scenarios, the aerosol-induced impact on soil moisture dominates the total NPP response (Table 3)." (Lines 626-629)

---

## Referee Report (RR1)

**Re-Review of Yue et al., ACPD**

**"Ozone and haze pollution weakens net primary productivity in China"**

I appreciate the authors' efforts to address the review comments. I would like to see a little more information on some of the points raised and I include some additional minor points below.

*3. The paper needs a more consistent time-scale. The overall results are presented as annual, however all the figures (except Fig 10) show summertime results only. The authors should either include evaluation for all seasons (or annual means), or present the final results only for summer. As is, the reader cannot judge model skill or response for other seasons.*

➔  The reason why our analyses and the Figures focus on the summer is that both GPP/NPP and air pollution (especially O3) reach maximum at this season. The largest interactions between carbon flux and air pollution are found for this season. It is not a contradiction to show Figures on the summer average and provide annual average impacts because the carbon loss in summer largely dominates the annual total. We found that, for O3 damages, "about 61% of such inhibition occurs in summer, when both photosynthesis and [O3] reach maximum of the year." (Lines 409-410). For the combined O3 and aerosol effects, "a dominant fraction (60% without AIE and 52% with AIE) of the reduced carbon uptake occurs in summer, when both NPP and [O3] reach maximum of the year." (Lines 474-476). We also elect to present and summarize the annual average results to the reader for consistency with regional carbon budget studies. Having the annual average values easily available facilitates comparison with other carbon flux impacts and carbon emissions. For example, we found that: "the combined effects of O3 and aerosols (Table 3) decrease total NPP in China by 0.39 (without AIE) to 0.80 Pg C yr-1 (with AIE), equivalent to 9-16% of the pollution-free NPP and 16-32% of the total anthropogenic carbon emissions". (Lines 469-471)

The authors themselves state that some of the results depend on the season (line 488) and this should be fully discussed. The authors may prefer to focus on summer in the main text Figures, but given that the impacts of other seasons are not negligible and the main conclusions are given as annual means, they MUST provide additional model evaluation in non-summer seasons in the Supplementary Materials. This should include evaluation of PM (AOD), O3, and radiation (Figures 1, 2, 3, 4). The results of this evaluation (consistency or not with summertime evaluation) should be briefly discussed in the main text.

*4. The paper should discuss the potential implications of the high bias in simulated diffuse fraction and potentially in O3 (the evaluation of simulated O3 is mixed).*

➔  We added following statements to discuss the implications of biases in diffuse fraction and O3: "Predicted [O3] is largely overestimated at urban sites but exhibits reasonable magnitude at rural sites (Figs 2 and 3). Measurements of background [O3] in China are limited both in space and time, restricting comprehensive validation of [O3] and the consequent estimate of O3 damages on the country level." (Lines 595-598)
"The model overestimates diffuse fraction in China (Fig. 4), likely because of simulated biases in clouds. Previously, we improved the prediction of diffuse fraction in China using observed cloud profiles for the region (Yue and Unger, 2017). Biases in simulated AOD and diffuse fraction introduce uncertainties in the aerosol DRF especially in the affected localized model grid cells. Yet, averaged over the China domain, our estimate of NPP change by aerosol DRF (0.09 Pg C yr-1) is consistent with the previous assessment in Yue and Unger (2017) (0.07 Pg C yr-1)." (Lines 619-625)

A follow-up question. As the authors emphasize that rural sites are more appropriate for evaluating their simulation, it seems reasonable to ask what fraction of the GPP change induced by pollution occurs over "urban" gridboxes? They suggest on line 541 that the change in GPP mainly derives from Eastern China, which is largely urban. This would provide some guidance as to how to interpret the relative urban vs rural $O_3$ simulation bias.

6. *Line 202-203: do these changes in biomass burning emissions seem realistic?*
→ The reviewer raises an interesting and provocative question. The future changes in biomass burning in China are small, and that is indeed realistic based on current understanding of fire activity in China today. For example, wildfire activity is limited in China today. We state in the manuscript: "Biomass burning emissions, adopted from the IPCC RCP8.5 scenario (van Vuuren et al., 2011), are considered as anthropogenic sources because most fire activities in China are due to human-managed prescribed burning. Compared with the GAINs inventory, present-day biomass burning is equivalent to <1% of the emissions for $NO_x$, $SO_2$, and $NH_3$, 1.6% for BC, 3.0% for CO, and 9.6% for OC." (Lines 213-217)

New sentence on fire activity in China being anthropogenic needs a literature reference.

8. *Lines 208-209: Please explain why isoprene emissions increase and monoterpene emissions decrease (text later indicates that land cover is fixed)*
→Please see above response to Point (7).

This is still a little unclear. Is the difference in emissions response for MT and ISOP emission to $CO_2$, T, GPP (are MT emissions sensitive to GPP?) due to very different sensitivities to these factors, or due to geographical factors (i.e. regions dominated by MT see larger changes in $CO_2$ than T, etc.)

12. *Section 3.1.2 & Figure 2: Please briefly discuss where the model is too high and too low and what species might contribute to these biases. Also quantify the last sentence (line 298-299)*
→We describe the AOD biases as follows: "Predicted AOD also reproduces the observed spatial pattern, but underestimates the high center in NCP by 24.6%." (Lines 339-340)
In the Discussion Section 4.2, we explain the cause of AOD biases: "Simulated surface $PM_{2.5}$ is reasonable but AOD is underestimated in the North China Plain (Fig. 2), likely because of the biases in aerosol optical parameters. Using a different set of optical parameters, we predicted much higher AOD that is closer to observations with the same aerosol vertical profile and particle compositions (Yue and Unger, 2017)." (Lines 615-619)
We revise the text as follows: "Evaluations at rural sites (Table S4), which represent the major domain of China, show a mean bias of -5% (Fig. 3). The magnitude of such bias is much lower than the value of 42.5% for the comparisons at urban-dominant sites (Fig. 2f)." (Lines 346-348)

Is there any particular aspect to the "different set of optical parameters" that improves the simulation (i.e. scattering, absorption, water uptake, etc.)? Why did the authors not then use these superior aerosol optical

parameters? A description & citation for current optical properties should be added to the Model Description.

**Additional Points**

1. Lines 340-341: It's not clear what this new text means. Was the baseline meteorology adjusted by these scaling factors? For each grid box? Please clarify/expand the description of this procedure.
2. Lines 360-366: line 360 indicates that NPP and GPP biases are less than 20%, but then specific biases of 23.7%, 20.6%, 40.0%, 51.2%, and 38.7% are not consistent with this. Please correct this text.
3. Section 3.1.3: The overestimate of diffuse fraction (line 398) seems likely to be associated with clouds (this is stated later in the text) given that aerosols are, if anything, underestimated. Have the authors compared the simulated clouds with other observational datasets? How do MERRA and the online clouds compare?
4. Figures 6, 8, 9, 10: could the authors indicate whether the local results are significant compared to interannual variability (as In Figure 7)

---

## Author Response (AR2)

**Re-Review of Yue et al., ACPD "Ozone and haze pollution weakens net primary productivity in China"**

We are grateful to the reviewer for their time and energy in providing helpful comments and guidance that have improved the manuscript. In this document, we describe how we have addressed the reviewer's comments. Referee comments are shown in black italics (first round) and light blue (second round). Author responses are shown in blue (first round) and magenta (second round) regular text.

I appreciate the authors' efforts to address the review comments. I would like to see a little more information on some of the points raised and I include some additional minor points below.

*3. The paper needs a more consistent time-scale. The overall results are presented as annual, however all the figures (except Fig 10) show summertime results only. The authors should either include evaluation for all seasons (or annual means), or present the final results only for summer. As is, the reader cannot judge model skill or response for other seasons.*

➔ The reason why our analyses and the Figures focus on the summer is that both GPP/NPP and air pollution (especially O3) reach maximum at this season. The largest interactions between carbon flux and air pollution are found for this season. It is not a contradiction to show Figures on the summer average and provide annual average impacts because the carbon loss in summer largely dominates the annual total. We found that, for O3 damages, "about 61% of such inhibition occurs in summer, when both photosynthesis and [O3] reach maximum of the year." (Lines 409-410). For the combined O3 and aerosol effects, "a dominant fraction (60% without AIE and 52% with AIE) of the reduced carbon uptake occurs in summer, when both NPP and [O3] reach maximum of the year." (Lines 474-476). We also elect to present and summarize the annual average results to the reader for consistency with regional carbon budget studies. Having the annual average values easily available facilitates comparison with other carbon flux impacts and carbon emissions. For example, we found that: "the combined effects of O3 and aerosols (Table 3) decrease total NPP in China by 0.39 (without AIE) to 0.80 Pg C yr-1 (with AIE), equivalent to 9-16% of the pollution-free NPP and 16-32% of the total anthropogenic carbon emissions". (Lines 469-471)

The authors themselves state that some of the results depend on the season (line 488) and this should be fully discussed. The authors may prefer to focus on summer in the main text Figures, but given that the impacts of other seasons are not negligible and the main conclusions are given as annual means, they MUST provide additional model evaluation in non-summer seasons in the Supplementary Materials. This should include evaluation of PM (AOD), O3, and radiation (Figures 1, 2, 3, 4). The results of this evaluation (consistency or not with summertime evaluation) should be briefly discussed in the main text.

➔ We added evaluation figures for annual means as suggested. The original Fig. S4 shows evaluations of annual carbon fluxes (corresponding to the summer results in Fig. 1). The newly added Fig. S5 shows evaluations of annual AOD, $[O_3]$, and $PM_{2.5}$ concentrations (corresponding to the summer results in Fig. 2). The original Fig. 3 has already included results in non-summer seasons. The newly added Fig. S6 shows evaluations of annual radiation and diffuse fraction (corresponding to the summer results in Fig. 4).

We have revised text properly to refer to these new figures. For example:

Evaluations at rural sites (Table S4) show a mean bias of -5% (Fig. 3). The magnitude of such bias is much lower than the values of comparisons at urban-dominant sites, where simulated $[O_3]$ is higher by 42.5% for the summer mean (Fig. 2f) and 55.6% for the annual mean (Fig. S5f). (Lines 354-357)

Simulated surface shortwave radiation agrees well with measurements at 106 sites for both summer (Figs 4a-4c) and annual (Figs S6a-S6c) means. (Lines 365-367)

Simulated diffuse fraction reproduces observed spatial pattern with high correlation coefficient ($r = 0.74$ for summer and $r = 0.65$ for annual, $p < 0.01$), though it is larger than observations on average by 25.2% in summer (Figs 4d-4f) and 35.2% for the annual mean (Figs S6d-S6f). (Lines 367-370)

4. *The paper should discuss the potential implications of the high bias in simulated diffuse fraction and potentially in O3 (the evaluation of simulated O3 is mixed).*

➔ We added following statements to discuss the implications of biases in diffuse fraction and O3: "Predicted [O3] is largely overestimated at urban sites but exhibits reasonable magnitude at rural sites (Figs 2 and 3). Measurements of background [O3] in China are limited both in space and time, restricting comprehensive validation of [O3] and the consequent estimate of O3 damages on the country level." (Lines 595-598)

"The model overestimates diffuse fraction in China (Fig. 4), likely because of simulated biases in clouds. Previously, we improved the prediction of diffuse fraction in China using observed cloud profiles for the region (Yue and Unger, 2017). Biases in simulated AOD and diffuse fraction introduce uncertainties in the aerosol DRF especially in the affected localized model grid cells. Yet, averaged over the China domain, our estimate of NPP change by aerosol DRF (0.09 Pg C yr-1) is consistent with the previous assessment in Yue and Unger (2017) (0.07 Pg C yr-1)." (Lines 619-625)

A follow-up question. As the authors emphasize that rural sites are more appropriate for evaluating their simulation, it seems reasonable to ask what fraction of the GPP change induced by pollution occurs over "urban" gridboxes? They suggest on line 541 that the change in GPP mainly derives from Eastern China, which is largely urban. This would provide some guidance as to how to interpret the relative urban vs rural O3 simulation bias.

→ The reviewer proposed an interesting question. For this study, we use grid resolution of 2°×2.5° latitude by longitude, which is too coarse to identify 'urban' gridboxes. However, we can answer the question based on information from 'China Statistical Yearbook for 2015' (see the table below). The total urban area in China is 184098.6 km$^2$, which is 2% of the total land area. Considering that most of cities are located in eastern part (>1/3 of domestic area), the percentage of urban area should be less than 6% in the east. As a result, we need to evaluate [O$_3$] at rural areas rather than urban areas, especially when rural concentrations are usually much higher.

In the text, we added the following statistics:

Based on 'China Statistical Yearbook for 2015' (http://www.stats.gov.cn), the total rural area accounts for >98% of the domestic area. Evaluations at rural sites (Table S4) show a mean bias of -5% (Fig. 3). (Lines 352-354)

**25-4 分地区城市建设情况（2014年）**

| 地 区 | 城区面积
（平方公里） | 建成区面积
（平方公里） | 城市建设
用地面积
（平方公里） | 本年征用
土地面积
（平方公里） | 城市人口
密 度
（人/平方公里） |
|---|---|---|---|---|---|
| 全 国 | 184098.6 | 49772.6 | 49982.7 | 1475.9 | 2419 |

*6. Line 202-203: do these changes in biomass burning emissions seem realistic?*

→The reviewer raises an interesting and provocative question. The future changes in biomass burning in China are small, and that is indeed realistic based on current understanding of fire activity in China today. For example, wildfire activity is limited in China today. We state in the manuscript: "Biomass burning emissions, adopted from the IPCC RCP8.5 scenario (van Vuuren et al., 2011), are considered as anthropogenic sources because most fire activities in China are due to human-managed prescribed burning. Compared with the GAINs inventory, present-day biomass burning is equivalent to <1% of the emissions for NOx, SO2, and NH3, 1.6% for BC, 3.0% for

CO, and 9.6% for OC." (Lines 213-217)

New sentence on fire activity in China being anthropogenic needs a literature reference.

→ We refer to the recent study by Zhou et al. (2017), which compiled a detailed biomass burning emission inventory in China and found that domestic straw burning, in-field straw burning, and firewood burning are the dominant biomass burning sources. In our revised paper, we added the reference as follows:

"Biomass burning emissions, adopted from the IPCC RCP8.5 scenario (van Vuuren et al., 2011), are considered as anthropogenic sources because most fire activities in China are due to human-managed prescribed burning (Zhou et al., 2017)." (Lines 216-218)

Zhou, Y., Xing, X., Lang, J., Chen, D., Cheng, S., Wei, L., Wei, X., and Liu, C.: A comprehensive biomass burning emission inventory with high spatial and temporal resolution in China, Atmospheric Chemistry and Physics, 17, 2839-2864, doi:10.5194/acp-17-2839-2017, 2017.

*8. Lines 208-209: Please explain why isoprene emissions increase and monoterpene emissions decrease (text later indicates that land cover is fixed)* →Please see above response to Point (7).

This is still a little unclear. Is the difference in emissions response for MT and ISOP emission to CO2, T, GPP (are MT emissions sensitive to GPP?) due to very different sensitivities to these factors, or due to geographical factors (i.e. regions dominated by MT see larger changes in CO2 than T, etc.)

→ Emissions of isoprene and monoterpene have different sensitivity to $CO_2$. Increases of $CO_2$ always inhibit monoterpene but not for isoprene, which is also dependent on photosynthesis that may increase due to $CO_2$ fertilization. In the revised text, we have explained this clearly:

"Leaf isoprene emissions are simulated using a biochemical model that depends on the electron transport-limited photosynthetic rate, intercellular CO2, canopy temperature, and atmospheric CO2 (Unger et al., 2013). Leaf monoterpene emissions depend on canopy temperature and atmospheric CO2 (Unger and Yue, 2014). Annual average isoprene emission in China increases by 5% (0.39 Tg C yr-1) between 2010 and 2030 in response to enhanced GPP and temperature that offset the effects of CO2-inhibition. Monoterpene emissions decrease by 5% (-0.25 Tg C) between 2010 and 2030 because CO2-inhibition outweighs the effects of increased warming." (Lines 235-242).

*12. Section 3.1.2 & Figure 2: Please briefly discuss where the model is too high and too low and what species might contribute to these biases. Also quantify the last sentence (line 298-299)* →We describe the AOD biases as follows: "Predicted AOD also reproduces the observed spatial pattern, but underestimates the high center in NCP by 24.6%." (Lines 339-340)

In the Discussion Section 4.2, we explain the cause of AOD biases: "Simulated surface PM2.5 is reasonable but AOD is underestimated in the North China Plain (Fig. 2), likely because of the biases in aerosol optical parameters. Using a different set of optical parameters, we predicted much higher AOD that is closer to observations with the same aerosol vertical profile and particle compositions (Yue and Unger, 2017)." (Lines 615- 619)

We revise the text as follows: "Evaluations at rural sites (Table S4), which represent the major domain of China, show a mean bias of -5% (Fig. 3). The magnitude of such bias is much lower than the value of 42.5% for the comparisons at urban-dominant sites (Fig.

2f)." (Lines 346-348)

Is there any particular aspect to the "different set of optical parameters" that improves the simulation (i.e. scattering, absorption, water uptake, etc.)? Why did the authors not then use these superior aerosol optical parameters? A description & citation for current optical properties should be added to the Model Description.

→ We use different optical parameters only because it is applied for a different radiation model. Optical parameters used by climate model NASA ModelE2-YIBs have been defined based on previous evaluations on global and regional scales. For this study, we cannot simply revise the parameters because this will affect the energy balance of climate model, resulting in possible overflow of calculation and/or incorrect climatic responses. In the revised paper, we added following description of optical properties:

"Size-dependent optical parameters of clouds and aerosols are computed from Mie scattering, ray tracing, and T-matrix theory, and include the effects of non-spherical particles for cirrus and dust (Schmidt et al., 2006)." (Lines 180-182).

**Additional Points**

1. Lines 340-341: It's not clear what this new text means. Was the baseline meteorology adjusted by these scaling factors? For each grid box? Please clarify/expand the description of this procedure.

   → We clarify as follows: "For these simulations, the month-to-month meteorological perturbations caused by aerosols are applied as scaling factors on the baseline forcing for each month at each grid square." (Lines 311-313)

2. Lines 360-366: line 360 indicates that NPP and GPP biases are less than 20%, but then specific biases of 23.7%, 20.6%, 40.0%, 51.2%, and 38.7% are not consistent with this. Please correct this text.

→ The bias of <21% is for the evaluations at national scale. As shown in Fig. 1 and Fig. S4, modeling biases are -15.8% for summer GPP, 20.6% for summer NPP, -3.9% for annual GPP, and 12.6% for annual NPP. The biases higher than 20% listed above are for regional scale. In the revised paper, we clarify as follows:

"Simulated GPP and NPP reproduce the observed spatial patterns with high correlation coefficients (R=0.46-0.86, $p < 0.001$) and relatively low model-to-observation biases ($\leq$ 21% on national scale)" (Lines 328-330).

3. Section 3.1.3: The overestimate of diffuse fraction (line 398) seems likely to be associated with clouds (this is stated later in the text) given that aerosols are, if anything, underestimated. Have the authors compared the simulated clouds with other observational datasets? How do MERRA and the online clouds compare?

→ The MERRA cloud fields are reanlayses data based on modeling and may be biased compared with satellite retrievals. Cloud variables and related radiation fields have been thoroughly evaluated in Schmidt et al. (2014). Compared with satellite data, cloud amount is biased by ±5% (which is reasonably low) over eastern China.

Schmidt, G. A., and coauthors: Configuration and assessment of the GISS ModelE2 contributions to the CMIP5 archive, Journal of Advances in Modeling Earth Systems, 6, 141-184, doi:10.1002/2013ms000265, 2014.

4. Figures 6, 8, 9, 10: could the authors indicate whether the local results are significant compared to interannual variability (as In Figure 7)

→ We show figures with significant tests in the revised paper. We added dots on Figure 6 to indicate grid squares with significant changes ($p < 0.05$). We replaced Figures 8 and 9

with new ones that only showing significant changes ($p < 0.05$). for Figure 10, the original plot shows only significant changes in (a)-(c) and we did not make further changes.

[revised manuscript text omitted]